



# The damaging character of shallow $20^{th}$ century earthquakes in the Hainaut coal area (Belgium)

Thierry Camelbeeck[1], Koen Van Noten[1], Thomas Lecocq[1], and Marc Hendrickx[1]

[1]Royal Observatory of Belgium, Seismology-Gravimetry. Avenue Circulaire 3, 1180 Uccle, Belgium.

**Correspondence:** Thierry Camelbeeck (Thierry.Camelbeeck@oma.be)

**Abstract.** Shallow, light to moderate magnitude earthquakes in stable continental regions can have a damaging impact on vulnerable surface constructions. In the coal area of the Hainaut province in Belgium, a century of shallow seismic activity occurred from the end of the $19^{th}$ century until the late $20^{th}$ century. This seismicity is the second largest source of seismic hazard in northwestern Europe, after the Lower Rhine Embayment. The present study synthesises the impact and damage caused by this unique shallow seismicity. Reviewing intensity data provided in official macroseismic surveys held by the Royal Observatory of Belgium, press reports, and contemporary scientific studies resulted in a complete macroseismic intensity dataset. The strong shaking of five seismic events with moment magnitudes ($M_W$) around 4.0, which occurred on 3 June 1911, 3 April 1949, 15 December 1965, 16 January 1966, and 28 March 1967, locally caused widespread moderate damage to buildings corresponding to maximum intensity VII in the EMS-98 scale. For 28 earthquakes, detailed macroseismic maps were created. Our study highlights the capability of shallow, small-magnitude earthquakes to generate damage. Subsequently, using the Hainaut intensity dataset, we modelled a new Hainaut intensity attenuation law and created relationships linking magnitude, epicentral intensity and focal depth. Using these relationships, we estimated the location and magnitude of pre-1985 earthquakes that occurred prior to deployment of the modern digital Belgian seismic network. Estimated focal depths allowed discriminating between two different types of earthquakes. Some events were very shallow, only a few hundred metres deep, suggesting a close link to mining activities. Other earthquakes, including the largest and most damaging events, occurred at depths greater than 2 km but no deeper than 6 km, which would exclude a direct relationship with mining, but yet still might imply a triggering causality. This work results in a new updated earthquake catalogue including 123 seismic events. Our attenuation modelling moreover suggests that current hazard maps overestimated ground motion levels in the Hainaut area due to the use of inadequate ground motion prediction equations. Our Hainaut attenuation model is hence useful to evaluate the potential impact of current and future, e.g. geothermal energy, projects in the Hainaut area and other regions with a similar geological configuration.

## 1 Introduction

In stable continental regions (SCR), 80% of the total seismic moment release occurs in the upper 7 km of the crust (Klose and Seeber, 2007). Hence, because of their shallow sources, moderate SCR earthquakes with magnitudes in the range of 4.0 to 6.0 are often more damaging in SCR than at plate boundaries (Camelbeeck et al., 2020). In Western Europe, this potential of





destruction of shallow SCR earthquakes was exemplified by the consequences of the 11 May 2011 Lorca (Spain) ($M_W$=5.1), 16 August 2012 Huizinge (The Netherlands) ($M_W$=3.6) and 11 November 2019 Le Teil (France) ($M_W$=4.9) earthquakes (Sira et al., 2019; Dost and Kraaijpoel, 2013; Association Française de génie Parasismique, 2011). Camelbeeck et al. (2014) highlight the potential danger of shallow small-magnitude earthquakes in stable Europe based on the observed damage caused by the

$M_W$=4 3/4 1884 Colchester (England) and $M_W$=4.6 1983 Liège (Belgium) earthquakes. The damaging impact of these events supports the need for considering shallow, small-magnitude earthquakes in seismic risk analyses of highly populated European low-seismicity regions, which is currently enhanced by the increase of induced seismicity by underground energetic resources (Nievas et al., 2020; Grigoli et al., 2017).

From 1887 to 1983, a century of significant seismicity occurred in the coal mining area of the Hainaut Province in Belgium

(Table 1). This seismicity is unique in Belgium and neighboring regions during this period as five events with $M_W$ 4.0 caused locally widespread, moderate to extensive damage to buildings. In southern Belgium, Namurian to Westphalian (Upper Carboniferous) coal seams have been intensively exploited in the $19^{th}$ and $20^{th}$ century in *"la bande Houillère"*, i.e. a narrow, 10 to 15 km wide geological region located between the Belgian cities of Mons in the West and Liège in the East (Fig. 1). This coal mining area is bordered in the south by the Midi Fault, which manifests the overthrusting of the Ardenne Allochton

(including the Dinant Fault-and-Thrust Belt and High-Ardenne slate belt) over the Brabant Parautochton. In the north, the coal mining area is limited up to the northern occurrence of the Westphalian (Fig. 2), which overlains the Lower Palaeozoic Brabant Massif. Mining in coal area in the province of Hainaut (further referred to as the Hainaut coal area) was focused on three basins: the Borinage-Mons basin, where up to more than 300 m Cretaceous to Cenozoic deposits cover the Westphalian, the Centre-La Louvière Basin and the Charleroi basin.

The main characteristics of seismic events in the Hainaut coal area are the high epicentral intensity and the rapid intensity decay with distance, suggesting shallow focal depths (Charlier, 1949; Van Gils, 1966; Ahorner, L., 1972; Van Gils and Zaczek, 1978). Despite the consequences of this "past" seismic activity there is no published synthesis and specific analysis about its impact and the damage caused by the different earthquakes. Providing an inventory of these effects and damage would be of great interest to identify the consequences of possible similar future activity, not only in the Hainaut area but also elsewhere in

Western Europe. Such an investigation is required for the analysis of the possible impact of deep geothermal projects that are currently in test phase or under development in the former Hainaut coal area (https://geothermiemons.be).

The Hainaut seismic activity is of great concern for seismic hazard assessment in the border area between France and Belgium. This is particularly of interest for the Eurocode-8 norm application in Belgian and French building regulations because the contribution of Hainaut seismic activity in these hazard maps is significant (Fig. 1) (Drouet et al., 2020; Leynaud et al.,

2001; Martin et al., 2002; Vanneste et al., 2014). For the current hazard maps, two different aspects of this seismicity deserve specific research. First, the origin of this seismicity stays unresolved and controversial (Descamps, 2009; Troch, 2018a). In hazard computations, natural seismicity would be a long-term stationary process, whereas seismicity induced by mining works would only be a past sporadic phenomenon. Hence, a reinterpretation on the origin would strongly modify its contribution to the seismic hazard. Second, in contrast to the observed strong intensity decay of these earthquakes, partly caused by the shal-
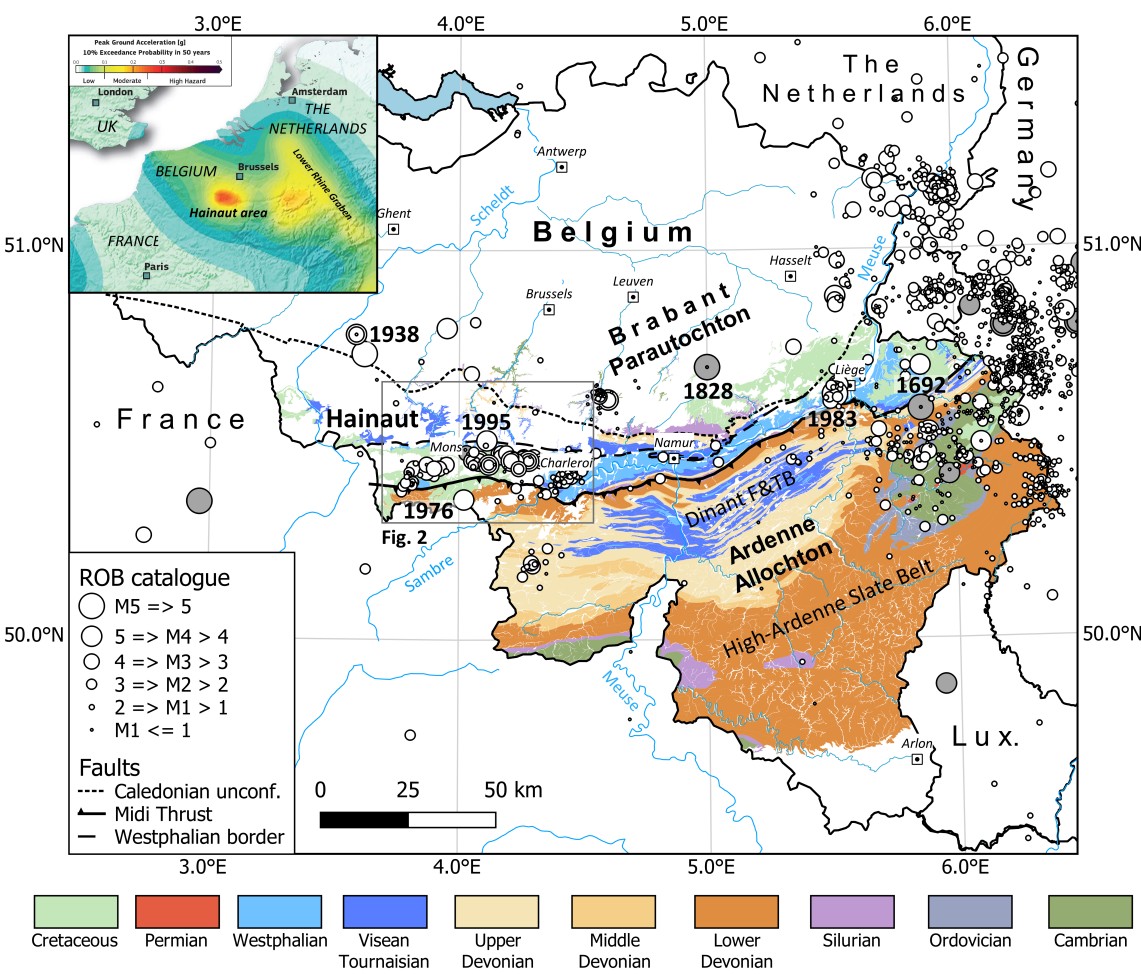

**Figure 1.** Regional seismicity and geological setting of the Hainaut coal area. Seismicity shown is the full seismic catalogue of the Royal Observatory of Belgium. Grey dots are historical earthquakes prior to the installation of the first seismometer in Belgium in 1911. The inset shows a zoom into the SHARE hazard map (Woessner et al., 2015) of the area around Belgium. Note the pronounced higher PGA exceedance in the Hainaut area based on the seismicity discussed in this paper. Geology in background based upon http://www.onegeology.org/. Reproduced with the permission of OneGeology. All rights Reserved.





lowness of the earthquake hypocentres, the influence area of the Hainaut seismicity seems too extended in the hazard maps. This inconsistency would result in the use of inappropriate ground motion prediction equations (GMPEs) in hazard assessment.

As most of the earthquake activity in the Hainaut coal area occurred before the implantation of a modern digital seismic network in Belgium, which started in 1985 (Camelbeeck et al., 1990), only the largest earthquakes since 1910 have been recorded by seismic stations. Smaller events are only known because they were reported by people and (or) caused slight

damage. Camelbeeck (1985a, b, 1993) and Camelbeeck et al. (1990) evaluated the magnitude of the largest events from seismic recordings. These studies underline the large uncertainties on earthquake locations from seismic phase measurements and conclude that for most events the centre of the area with the largest observed intensity would better correspond to the real epicentre than the location obtained from arrival time measurements. Due to the uncertainty on focal depths, instrumental evaluations were only able to suggest that they would certainly not exceed 7-8 km (Camelbeeck, 1990) and, to date, more

accurate depth estimations are lacking. Macroseismic data are, however, a good and the only alternative to determine earthquake source parameters and tackle the context of this seismicity and related seismic hazard issues.

In this paper, we collected all available macroseismic data of this unique seismicity and searched for additional information providing a complete macroseismic dataset of sufficient quality to answer the questions that the Hainaut seismicity raised. We use this dataset to properly estimate both the impact of this seismicity and the way intensity attenuates with distance,

which allows improving epicentre and depth determination, and opening up the path to intensity modelling in seismic hazard assessment. First, we briefly present the earthquake catalogue and the sources of information available on this seismicity. Second, we chronologically summarise the effects and damage of the largest events and provide macroseismic maps for them. Maps and sources are structured in an Atlas presentation in the Supplement (further referred to as the Atlas) and associated communal intensity data points (IDPs) are provided for each earthquake in the Supplement. Then, we develop a regional

intensity attenuation relationship valid for the Hainaut coal area, which allows better estimating the earthquake focal depth from the intensity dataset. We also provide a predictive model of earthquake magnitude related to epicentre intensity and focal depth. Finally, we discuss the benefit of our study in the perspective of current seismic hazard studies, and provide a new earthquake catalogue of the Hainaut seismicity from 1887 to 1983 mainly constrained on macroseismic data.

## 2 Earthquake catalogue

The Hainaut earthquakes are included in the earthquake catalogue maintained by the Royal Observatory of Belgium (ROB; see Data Availability).

To create the earthquake catalogue, Camelbeeck (1993) initially reviewed all the recordings of seismic stations in Belgium and neighbouring countries that could have reported phase arrival times and amplitude measurements for earthquakes in Belgium. Between 1898 and 1958, the only seismic station in Belgium was Uccle (Brussels). Its capability to record local

earthquakes was operational from 1909 onwards. However, the station was only sensitive enough to detect the largest earthquakes, and numerous felt earthquakes were too small to leave a trace on the black smoked or photo paper recordings. Hence, the ROB catalogue was extended by including felt Hainaut earthquakes that were not recorded by seismic instruments before







**Figure 2.** Geological setting of the 1887-1983 seismicity in the Hainaut province with local geological map as background. Borinage-Mons basin, La Louvière Basin and the Centrum Basin (Charleroi area) are the main coal regions in the Hainaut province. Seismicity (up to 2020) coloured in function of time and sized to magnitude. Black error bars show location uncertainty. Numbers next to the largest earthquakes refer to events in Table 1 and to macroseismic maps in the Atlas in the Supplement. Geology in background based upon http://www.onegeology.org/. Reproduced with the permission of OneGeology. All rights Reserved.





1958. However, their reporting is not homogeneous during this period. For the period between 1896 and 1936, Somville (1936)
established a list including some events that were not recorded in Uccle but that were reported in press reports, in communica-
tions from local collieries or by local correspondents. The catalogue also contains non-instrumentally recorded aftershocks of
the April 1949 Havré earthquakes reported in the press, and 12 earthquakes that occurred in the fifties and that were listed in
the Belgian activity reports of the International Union of Geodesy and Geophysics (IUGG).

After 1958 and up to 1985, adding a few additional stations slightly improved the seismic monitoring in Belgium (Camel-
beeck, 1985a). The higher sensitivity of the seismometers at the permanent stations in Dourbes and Membach, operating
respectively since 1958 and 1977, allowed detecting smaller, even not felt, seismic events. Hence, from 1958, the bulletin of
Belgian seismic stations includes all the potentially felt events. After 1985, the installation of a modern digital seismic network
allowed the detection and precise location of $M_L$>1.0 earthquakes in the Hainaut area (Lecocq et al., 2013). By the exception
of weakly felt earthquakes in 1987 in the Dour area (Camelbeeck, 1988), no more events were sufficiently strong to be felt
and the seismicity stayed at a very low level in Hainaut. Let's note that the $M_W$=4.1 earthquake that occurred on 20 June 1995
had its epicentre near Le Roeulx just north of the coal area (Figs. 1 and 2). With a focal depth of 25 km, the hypocentre was
located in the lower crust of the Brabant Massif. It was felt on a large part of the Belgian territory and in northern France with
an epicentral intensity of V (Fig. S31 in the Supplement).

Initially, we started our study using the list of Hainaut earthquakes reported in the ROB catalogue, but the new knowledge
acquired in this study allowed us to complete and improve the location reliability and to evaluate the magnitude for all events.
This resulted in an updated catalogue of 123 Hainaut earthquakes between 1887 and 1985 that is now fully integrated in the
ROB catalogue. Table 1 reports the 28 events for which we provide a macroseismic map, numbered with a leading $S$ in the
Supplement (see Supplement and Data Availability).

## 3   Macroseismic information and intensity evaluation

### 3.1   Sources of information

Our study is based on macroseismic information that is derived from various sources, including published scientific works
contemporaneous with the earthquakes, the official macroseismic survey of the Royal Observatory of Belgium, press reports,
letters to the ROB, and ROB, collieries company and administration reports. A detailed overview of these sources is provided
in the Atlas in the Supplement.

Scientific studies have described the effects and (or) damage caused by the Hainaut earthquakes in large detail (de Munck,
1887; Camelbeeck et al., 2021; Marlière, 1951; Cornet, 1911; Cambier, 1911; Capiau, 1920; Charlier, 1949; Van Gils, 1966).
Some works contain the own observations of the author(s), complemented by testimonies collected by interviewing local
people, similar as today's Macroseismic Intervention Group (Sira, 2015) would do.

The official ROB macroseismic survey (since 1932) is an indispensable tool to map the earthquake's effects and to estimate
magnitude, focal depth, epicentral intensity decrease and its relation with the local geological subsurface (Cecić and Musson,
2004). Between 3 April 1949 and 9 August 1983 19 official ROB surveys were organised in Hainaut. 17 of them were usable





**Table 1.** *Parameter info of 28 Hainaut coal area earthquakes that have sufficient macroseismic data to be mapped (see Atlas). See Supplement for complete explanation of all catalogue parameters.* **Map:** *Atlas map number;* **idE:** *ROB catalogue number.* **Inq:** *event with official ROB macroseismic inquiry;* **METHOD:** *method to compute macroseismic epicentre (G. Imax(-1): geocentre of the IDPs with Imax and Imax-1 intensities; G. Perc.: geocentre of all the IDPs);* **ERRH:** *Uncertainty on the reported epicentre in km;* **DEPTH:** *focal depth (km) estimated from the intensity attenuation modelling, depths in brackets are estimated from Imax;* **ERRZ:** *Focal depths (in km) using the Hainaut intensity attenuation law. Focal depths inside brackets are estimated from Imax;* $M_W$ _m: *Equivalent $M_W$ determined from macroseismic data using the empirical relationships developed in this study;* **IMAX:** *maximum observed intensity;* **PERC.:** *Radius of perceptibility of the seismic event in km. R3: Radius of intensity III, R4: Radius of intensity IV;* **ERRM:** *Uncertainty on estimated magnitude;* **IDPs:** *Number of IDPs.*

| MAP | ID_E | DATE | TIME | REGION | LAT | LON | METHOD | ERRH | DEPTH | ERRZ | $M_L$ | MS | $M_W$_m | $M_W$ | IMAX | PERC. | ERRM | IDPs |
|---|---|---|---|---|---|---|---|---|---|---|---|---|---|---|---|---|---|---|
| S1 | 449 | 1911-04-12 | 16:15:–.– | CUESMES | 50.44 | 3.92 | G. Imax(-1) | 1.8 | [2.4] | [1.1] | | | | 3.1 | 4 | 5.4 R3 | 0.5 mac | 22 |
| S2 | 465 | 1911-06-01 | 22:51:58.– | RANSART | 50.45 | 4.46 | G. Imax(-1) | 1.9 | 4.3 | 1.8 | 4.2 | 3.8 | | [3.9] | 6 | 13.5 R4 | 0.3 M | 53 |
| S3 | 466 | 1911-06-03 | 14:35:54.– | GOSSELIES | 50.46 | 4.45 | G. Imax(-1) | 0.6 | [1.4] | [0.7] | 4.4 | | | [4.0] | 7 | 7.7 R4 | 0.3 M | 16 |
| S4 | 476 | 1920-01-17 | 03:11:04.– | HORNU | 50.44 | 3.82 | G. Imax(-1) | 0.8 | [1.6] | [0.5] | 3.7 | | | [3.5] | 6 | 5.3 R3 | 0.3 M | 12 |
| S5 | 488 | 1931-05-09 | 12:25:56.– | HOUDENG-AIMERIES | 50.48 | 4.15 | G. Perc. | 0.9 | [0.6] | [0.2] | 2.8 | | | [3.0] | 4_5 | 2.5 R3 | 0.3 M | 5 |
| S6 | 505 | 1936-11-05 | 00:41:44.– | GOUY-LEZ-PIETON | 50.47 | 4.3 | G. Perc. | 0.9 | [2.2] | [0.9] | | | | 3.3 | 4_5 | 3.4 R4 | 0.6 mac | 5 |
| S7 | 517 | 1940-01-07 | 16:28:52.– | LA LOUVIERE | 50.47 | 4.17 | G. Imax(-1) | 0.3 | [1.5] | [0.6] | | | | 3.5 | 5 | 5.6 R3 | 0.5 mac | 17 |
| S8 | 518 | 1940-01-07 | 20:32:44.– | LA LOUVIERE | 50.47 | 4.2 | G. Imax(-1) | 1.9 | | | | | | 3.1 | 4 | 4.4 R3 | 0.5 mac | 7 |
| S9 | 519 | 1940-01-09 | 03:42:07.– | LA LOUVIERE | 50.48 | 4.17 | G. Imax(-1) | 0.2 | [2.8] | [1.4] | | | | 3.3 | 4_5 | 7.6 R3 | 0.5 mac | 10 |
| S10 | 534$^{inq}$ | 1949-04-03 | 12:33:40.– | HAVRE-BOUSSOIT | 50.46 | 4.08 | G. Imax(-1) | 1.8 | 2.2 | 0.8 | 4.6 | 4.3 | | [4.1] | 7 | 18.0 R3 | 0.3 M | 134 |
| S11 | 538 | 1949-04-14 | 01:09:14.– | HAVRE-BOUSSOIT | 50.46 | 4.07 | G. Imax(-1) | 3 | [3.7] | [1.6] | | | | 3.5 | 5 | 8.5 R3 | 0.5 mac | 15 |
| S12 | 539 | 1949-04-14 | 05:12:21.– | HAVRE | 50.46 | 4.06 | G. Imax(-1) | 1.6 | [2.4] | [1.5] | 3.8 | | | [3.6] | 6 | 9.5 R3 | 0.3 M | 21 |
| S13 | 547$^{inq}$ | 1952-10-21 | 21:15:–.– | QUAREGNON | 50.43 | 3.88 | G. Imax(-1) | 2.2 | [2.9] | [1.9] | | | | 3.1 | 4 | 5.5 R3 | 0.5 mac | 21 |
| S14 | 548$^{inq}$ | 1952-10-22 | 07:–:–.– | FRAMERIES | 50.42 | 3.9 | G. Imax(-1) | 0.8 | [3.0] | [1.0] | | | | 2.8 | 3 | 3.5 R3 | 0.4 mac | 11 |
| S15 | 549$^{inq}$ | 1952-10-27 | 06:11:–.– | QUAREGNON | 50.43 | 3.87 | G. Imax(-1) | 2 | 3.5 | 1.2 | | | | 3.5 | 5 | 11.1 R3 | 0.5 mac | 45 |
| S16 | 562$^{inq}$ | 1954-07-10 | 17:18:21.– | FLENU | 50.44 | 3.9 | G. Imax(-1) | 1.5 | 3.3 | 1.2 | | | | 3.5 | 5 | 8.8 R3 | 0.5 mac | 44 |
| S17 | 582$^{inq}$ | 1965-12-15 | 12:07:15 | STREPY-BRACQUEGNIES | 50.45 | 4.12 | G. Imax(-1) | 0.5 | 2.7 | 0.8 | 4.4 | | | 4 | 7 | 20.7 R3 | 0.3 M | 99 |
| S18 | 587$^{inq}$ | 1966-01-16 | 00:13:19 | MORLANWELZ-MARIEMONT | 50.46 | 4.24 | G. Imax(-1) | 1.7 | [2.6] | [1.4] | 2.7 | | | [2.9] | 4 | 7.2 R3 | 0.3 M | 25 |
| S19 | 588$^{inq}$ | 1966-01-16 | 06:51:34 | MORLANWELZ-MARIEMONT | 50.47 | 4.26 | G. Imax(-1) | 1.8 | 3.3 | 1.6 | 3.8 | | | 3.5 | 5 | 8.5 R3 | 0.3 M | 41 |
| S20 | 589$^{inq}$ | 1966-01-16 | 12:32:50 | MORLANWELZ-MARIEMONT | 50.46 | 4.26 | G. Imax(-1) | 0.6 | 2.1 | 0.9 | 4.4 | | | 4 | 7 | 24.9 R3 | 0.3 M | 120 |
| S21 | 597$^{inq}$ | 1967-03-28 | 15:49:25 | CARNIERES | 50.46 | 4.28 | G. Imax(-1) | 1.3 | 3 | 1 | 4.5 | | | 4.1 | 7 | 29.3 R3 | 0.3 M | 143 |
| S22 | 603$^{inq}$ | 1968-08-12 | 07:26:41 | LA LOUVIERE | 50.46 | 4.21 | G. Imax(-1) | 1.7 | 2.3 | 1 | 3.7 | | | 3.6 | 5 | 6.7 R3 | 0.3 M | 29 |
| S23 | 606$^{inq}$ | 1968-08-13 | 16:57:14 | LA LOUVIERE | 50.46 | 4.21 | G. Imax(-1) | 2 | 2.3 | 0.8 | 4.1 | | | 3.9 | 6 | 11.5 R3 | 0.3 M | 59 |
| S24 | 607$^{inq}$ | 1968-09-23 | 04:08:13 | MORLANWELZ-MARIEMONT | 50.46 | 4.23 | G. Imax(-1) | 2 | 2.8 | 1.7 | 3 | | | 3.2 | 5 | 6.2 R3 | 0.3 M | 25 |
| S25 | 608$^{inq}$ | 1968-09-23 | 05:47:16 | HAINE-SAINT-PIERRE | 50.47 | 4.22 | G. Imax(-1) | 1.2 | [2.4] | [1.1] | 2.9 | | | 3 | 4 | 4.7 R3 | 0.3 M | 25 |
| S26 | 612$^{inq}$ | 1970-11-03 | 08:46:00 | MARCHIENNE-AU-PONT | 50.41 | 4.41 | G. Imax(-1) | 1.6 | 2.3 | 1 | 3.9 | | | 3.6 | 5 | 9.8 R3 | 0.3 M | 31 |
| S27 | 627$^{inq}$ | 1976-10-24 | 20:33:28 | GIVRY | 50.36 | 4.02 | G. Imax(-1) | 2.4 | 5.5 | 1.7 | 4.2 | | | [3.9] | 6 | 16.0 R3 | 0.3 M | 95 |
| S28 | 641$^{inq}$ | 1982-09-14 | 19:24:35 | CARNIERES | 50.44 | 4.24 | G. Imax(-1) | 2 | [3.5] | [1.6] | 3.4 | | | [3.4] | 4 | 6.9 R3 | 0.3 M | 18 |

to evaluate intensity (indicated with $^{inq}$ in Table 1) in the EMS-98 macroseismic scale (Grünthal et al., 1998) and to compose a macroseismic map (see Atlas). In the Supplement, for each of these 17 earthquakes, an inquiry book is provided that presents an English translation of the reply to the ROB questionnaire and provides the minimum (Imin) and maximum intensity (Imax) for each locality. Intensities gathered from official forms provided convincing results because municipalities in Belgium were

small (mean area size of only 19 km$^2$) and numerous (2359 communes). After the big community fusion in 1977, in which Belgium changed from 2359 to 596 communities (with mean area size of 82 km$^2$), macroseismic surveys of more recent earthquakes lost the quality and resolution they once had because the new communities cover a too large area to be represented by only one intensity value.





At the end of the $19^{th}$ century and beginning of the $20^{th}$ century, local and regional press reports were very beneficial
documents for seismologists to summarise an earthquake's impact (Alexandre et al., 2007; Camelbeeck et al., 2021). We
consulted the ROB database, La Louvière record-office collections and scanned press archives of the State Archives of Belgium
State Archives of Belgium (2021) to extend our Hainaut earthquake information. The list of consulted newspapers is presented
in the Atlas.

Additional information comes from letters of individuals or small reports addressed by the collieries companies to the ROB at
the time of the mining exploitation (Somville, 1936). The ROB also organised field missions after some earthquakes, providing
reports of the observations.

### 3.2 Intensity evaluation

We evaluated local intensities for each earthquake at each locality for which information is sufficient. Intensity is determined
in the EMS-98 scale, the current standard in Europe. Its great advantage is the use of building vulnerability classes allowing
to integrate the current state of the building stock in the intensity determination (Grünthal et al., 1998). For the earthquakes
that occurred during the first half of the $20^{th}$ century, we based our analysis on the macroseismic information explained in
the sections above. For the Hainaut earthquakes after 1949, we assessed intensity mainly from ROB official surveys. The
background how we evaluated building vulnerability and assessed intensity from these various sources and damage reports, is
explained in detail in Appendix A.

### 3.3 The Hainaut intensity dataset

Carefully evaluating all the intensity data sources lead us to compose the Hainaut intensity dataset. Table 2 presents the sum
of IDPs that reached a certain intensity value, which is the mean of Imin and Imax values, representing the range of possible
intensity values deduced from the macroseismic information.

## 4 Description of the strongest, often damaging, Hainaut earthquakes

In this section, we chronologically present information on the earthquakes that were widely felt or caused damage in the
Hainaut coal area (reported in Table 1). As newspapers often report precise addresses or places in cities where some specific
damage occurred, we geocoded this information and located the type of damage on the macroseismic maps in the paper and in
the Atlas.

### 4.1 The March-June 1911 Ransart – Gosselies seismic sequence

The first known earthquake that caused damage in the Hainaut coal area occurred at 0h05m on 29 March 1911 north of the
city of Charleroi. A violent tremor accompanied by a tremendous noise awakened the population of the communes of Ransart,
Gosselies, Heppignies and Wayaux. It shook the houses for a few seconds, enough to knock over furniture, break dishes, open
unlocked doors and frighten people. However, as the earthquake occurred at midnight, there was no notice of the event outside





**Table 2.** Summary of intensity (EMS-98) data for the largest earthquakes in the Hainaut coal area and which mapped in the Atlas. Map: map number in Supplement. Total IDPs: Amount of IDPS with mean intensity of Imin and Imax. Inq: earthquake with an official ROB intensity survey; *: earthquake used for Hainaut intensity attenuation modelling. Geology in background based upon http://www.onegeology.org/, with the permission of OneGeology.

| Map | id_earth | Date | F | II | II-III | III | III-IV | IV | IV-V | V | V-VI | VI | VI-VII | VII | Total |
|---|---|---|---|---|---|---|---|---|---|---|---|---|---|---|---|
| S1 | 449 | 1911-04-12 | | 2 | 3 | 14 | 2 | 1 | | | | | | | **22** |
| S2 | 465* | 1911-06-01 | | | | 2 | | 31 | | 14 | 2 | 4 | | | **53** |
| S3 | 466 | 1911-06-03 | 11 | | | | | 2 | | 1 | | 1 | | 1 | **16** |
| S4 | 476 | 1920-01-17 | 9 | | | | | | | 1 | | 2 | | | **12** |
| S5 | 488 | 1931-05-09 | 4 | | | | | | 1 | | | | | | **5** |
| S6 | 505 | 1936-11-05 | | | | | | | 5 | | | | | | **5** |
| S7 | 517 | 1940-01-07 | | 2 | 6 | 5 | 3 | | | 1 | | | | | **17** |
| S8 | 518 | 1940-01-07 | 7 | | | | | | | | | | | | **7** |
| S9 | 519 | 1940-01-09 | 5 | | | 2 | | 2 | 1 | | | | | | **10** |
| S10 | 534$^{inq,*}$ | 1949-04-03 | | 24 | 3 | 36 | 1 | 32 | 8 | 13 | 6 | 7 | 2 | 2 | **134** |
| S11 | 538 | 1949-04-14 | 7 | | | | | 6 | | 2 | | | | | **15** |
| S12 | 539 | 1949-04-14 | 12 | | | | | | 2 | 3 | 2 | 2 | | | **21** |
| S13 | 547$^{inq}$ | 1952-10-21 | | 2 | 1 | 12 | 2 | 4 | | | | | | | **21** |
| S14 | 548$^{inq}$ | 1952-10-22 | | 1 | 1 | 7 | 1 | 1 | | | | | | | **11** |
| S15 | 549$^{inq}$ | 1952-10-27 | | 6 | | 13 | 4 | 12 | 2 | 8 | | | | | **45** |
| S16 | 562$^{inq,*}$ | '1954-07-10 | | 11 | | 7 | 1 | 9 | 2 | 12 | 2 | | | | **44** |
| S17 | 582$^{inq,*}$ | 1965-12-15 | | 23 | | 30 | 6 | 17 | | 19 | | 2 | | 2 | **99** |
| S18 | 587$^{inq}$ | 1966-01-16 | | 3 | 1 | 13 | 4 | 2 | 2 | | | | | | **25** |
| S19 | 588$^{inq,*}$ | 1966-01-16 | | 15 | 1 | 8 | 2 | 12 | 1 | 2 | | | | | **41** |
| S20 | 589$^{inq,*}$ | 1966-01-16 | | 37 | | 42 | 2 | 22 | 1 | 12 | | 3 | | 1 | **120** |
| S21 | 597$^{inq,*}$ | 1967-03-28 | | 40 | | 56 | 3 | 22 | 1 | 10 | | 9 | | 2 | **143** |
| S22 | 603$^{inq,*}$ | 1968-08-12 | | 6 | | 2 | 1 | 12 | | 8 | | | | | **29** |
| S23 | 606$^{inq,*}$ | 1968-08-13 | | 18 | | 9 | | 10 | | 17 | 1 | 4 | | | **59** |
| S24 | 607$^{inq,*}$ | 1968-09-23 | | 10 | | | 4 | 9 | 1 | 1 | | | | | **25** |
| S25 | 608$^{inq}$ | 1968-09-23 | | 13 | | 5 | 2 | 5 | | | | | | | **25** |
| S26 | 612$^{inq,*}$ | 1970-11-03 | | 6 | | 9 | 3 | 5 | | 8 | | | | | **31** |
| S27 | 627$^{inq}$ | 1976-10-24 | | 24 | | 24 | 1 | 33 | 1 | 10 | | 2 | | | **95** |
| S28 | 641$^{inq}$ | 1982-09-14 | | 1 | 1 | 8 | 1 | 7 | | | | | | | **18** |
| **Total Intensity** | | | **55** | **242** | **13** | **305** | **45** | **259** | **28** | **142** | **13** | **36** | **2** | **8** | **1148** |

a radius of 3 to 4 km from the barycentre of all macroseismic data points. In Ransart, many cracks in houses were reported and the school chimney was knocked over. The magnitude of this seismic event is estimated to M$_W$=3.1 from the seismic





recordings at the Uccle seismic station located nearly 40 km north of the assumed epicentre. After this earthquake, some light tremors occurred on 12 April 1911 (Fig. S1), in the region of Mons and Cuesmes on the other side of the coal mining area.

Two months later, the earth shook again north of Charleroi, but more strongly with a $M_W$=3.9 event on 1 June at 22h51m (Figs. 3 and S2) and a $M_W$=4.0 event on 3 June at 14h35m (Fig. S3). The epicentral area of the 1 June 1911 earthquake includes
the localities of Gosselies, Lambusart and Ransart where the tremors were violent enough to awaken most of the inhabitants, knocking down many chimneys and causing cracks in the least resistant buildings (Cambier, 1911). According to newspapers *"Le courrier de l'Escaut – 4/6/1911"* and *"La Meuse – 3/6/1911"* the most affected locality was Ransart where more or less 50 chimneys collapsed and a parked train was thrown off the tracks. A wire-drawing factory would have collapsed in Gosselies, killing 1 person and injuring 3 others. We assessed intensity to VI in Ransart, Gosselies and Lambusart. In the neighboring
localities of Roux and Courcelles, the visible damage was limited to a few smokestacks that were knocked down (intensity V-VI).

Curiously, Cambier (1911) did not provide any information on the 3 June 1911 earthquake, which was more damaging than the 1 June 1911 earthquake as reported by the newspapers. In Gosselies, there were entire streets where almost all the chimneys were knocked over, damaging roofs and skylights. In the houses, objects hanging from the walls were thrown to the
ground (*"Journal de Bruxelles – 5/6/1911"*). *"La Gazette de Charleroi – 4/6/1911"* mentions that many houses are cracked and windows are broken, and the damage would be more concentrated near the Gosselies station. The importance of the damage led us to estimate intensity to VII in Gosselies. Newspapers also describe damage in Ransart, but they are less important than during the 1 June 1911 earthquake. The damage repartition clearly suggests that the earthquake of 3 June would be located in Gosselies, 2-3 km to the northwest of the epicentre of the 1 June seismic event in Ransart.

**4.2 The 17 January 1920 earthquake in the Borinage**

This $M_W$=3.5 earthquake (Fig. S4) recorded by the seismic station of Uccle occurred at 3h11m in the morning. The newspapers report that falling chimneys tore off rooftiles in Boussu and Hornu. In the miners' houses, objects collided with each other, were moved or knocked over. Capiau (1920) published a brief notice of his observations on the earthquake effects. Maximum intensity is set to VI based on these newspaper reports.

**4.3 The 9 May 1931 earthquake east of La Louvière**

This $M_W$=3.0 event is a smaller event than the previous ones. In the epicentral area, many residents rushed outside while in some neighbourhoods the doors of the houses opened. A chimney collapsed in Houdeng-Aimeries (Fig. S5).

**4.4 The 5 November 1936 Trazegnies-Chapelle earthquake**

This $M_W$=3.3 earthquake (Fig. S6) did not cause any damage but many inhabitants of the communes of Trazegnies, Piéton,
Gouvy-Lez-Piéton, Godarville and Chapelle were awakened by the shakings. The main observations are that windows vibrated while small objects were knocked over from shelving furniture and fireplaces (*"L'Indépendance Belge - 7/11/1936"*).



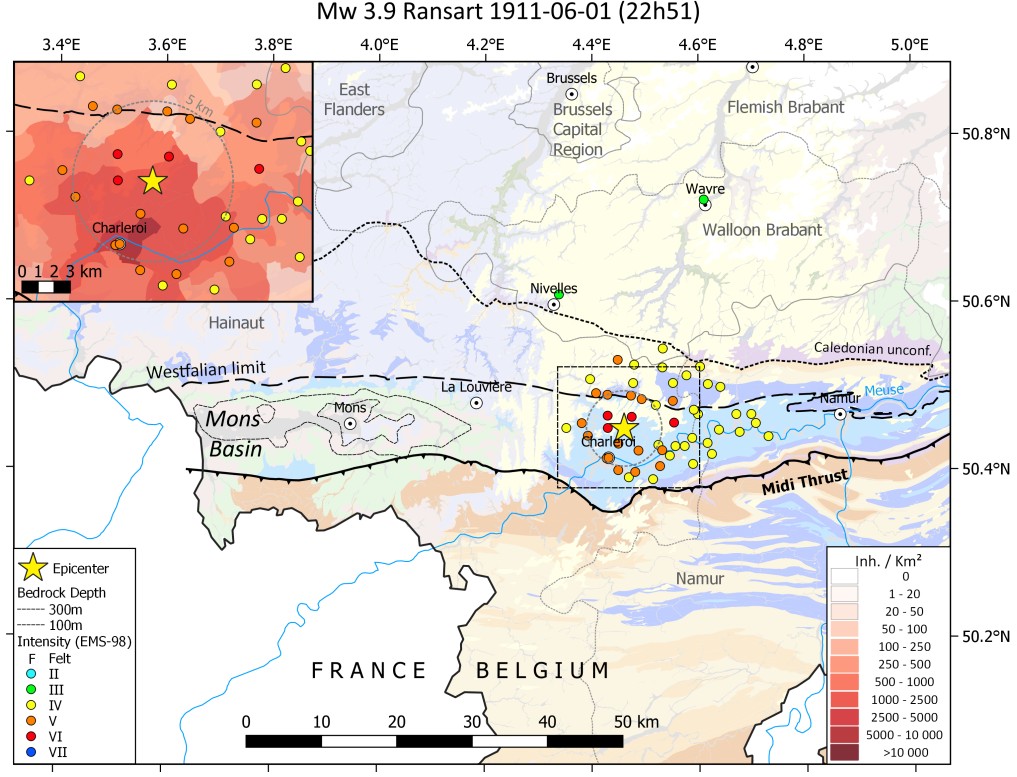

**Figure 3.** Macroseismic map of the 1 June 1911 $M_W$=3.9 ($M_L$=4.2) Ransart earthquake (nr 2 in Table 1). Maximal intensity = VI. Geology in background based upon http://www.onegeology.org/. Reproduced with the permission of OneGeology. All rights Reserved. The inlet shows the population density.

### 4.5 The 7 and 9 January 1940 earthquakes east of La Louvière

Three small events recorded by the Uccle seismic station occurred in January 1940 near La Louvière (Figs. S7, S8, S9). The first of $M_W$=3.5 on 7 January at 16h28m was best recorded in Uccle and was the most violent of the sequence. In La Louvière,
furniture was moved while vases placed on the marbles of the fireplaces as well as doors and windows shook. The newspaper "La Gazette de Charleroi" shows a photo of a damaged fireplace in Saint-Vaast indicating that slight damage was observed. The two earthquakes that followed were more weakly felt. The 9 January 1940 earthquake ($M_W$=3.1) that occurred early in the morning woke up few people but has been locally felt by workers in the coal mines near La Louvière (Fig. S9).

### 4.6 The 3 and 14 April 1949 Havré-Boussoit earthquakes

One of the strongest earthquakes in the Hainaut coal area occurred on 3 April 1949 at 12h33m in the region of Havré, 8 km to the east of Mons. This $M_W$=4.1 earthquake was preceded at 12h27m by a $M_W$=3.7 event, which was also strongly felt. The ROB conducted a detailed survey about the damage and effects caused by the 12h33 earthquake (Charlier, 1951). This earthquake is



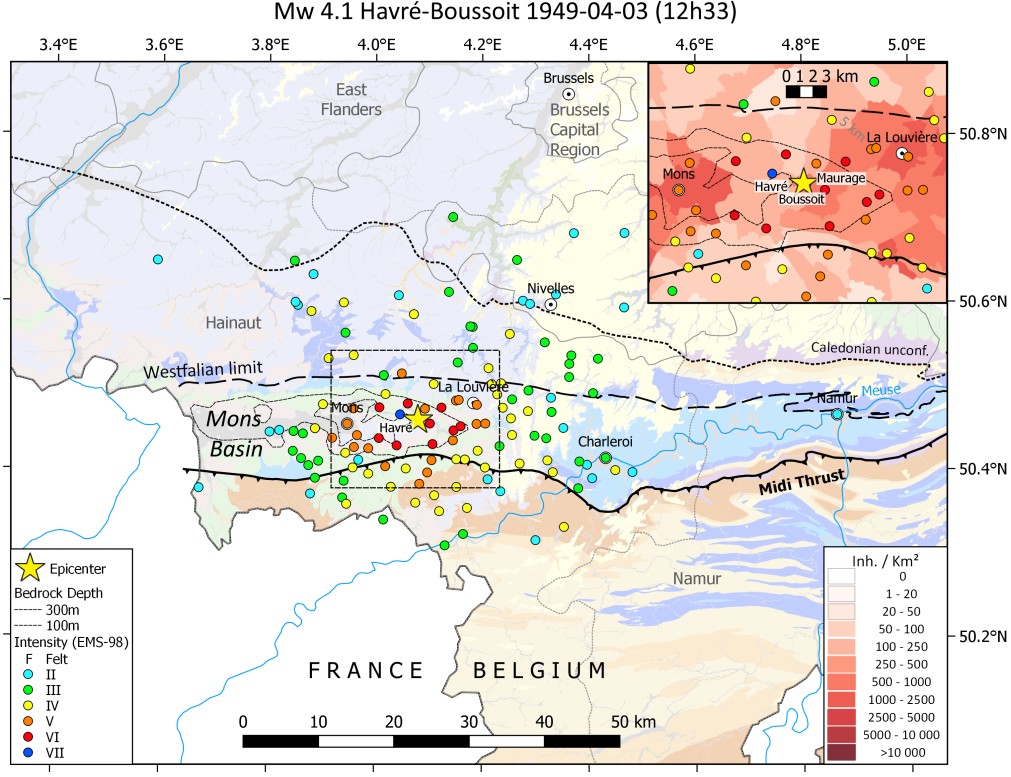

**Figure 4.** Macroseismic map of the 3 April 1949 $M_W$=4.1 ($M_L$=4.6) earthquake in the Borinage (nr 10 in Table 1). Maximal intensity = VII. Geology in background based upon http://www.onegeology.org/. Reproduced with the permission of OneGeology. All rights Reserved. The inlet shows the population density.

the first one for which the ROB organised an official survey on a large part of the Belgian territory. The macroseismic map based on our reassessment of intensities in the EMS-98 macroseismic scale are reported on Figures 4 and S10. The most affected localities are Boussoit, Havré and Maurage where we estimate intensity to VII. In Havré, there was a beginning of panic after the 12h33 tremor, which was so violent that more than 80% of the chimneys out of 1400 dwellings were disrupted, of which 50% needed to be completely rebuilt, and 150 had been completely overturned. In Boussoit, at least 70% of the chimneys were damaged or collapsed, while in Maurage about 200 and 25 chimneys were respectively damaged and overturned. In Maurage, the vault of the church choir was damaged by a crack while in Trivières, a slag heap has collapsed, endangering the neighbouring dwellings. The earthquake was followed by a number of aftershocks that were felt in the epicentral area. Only few of them were recorded at the Uccle seismic station and (or) reported in newspapers with sufficient precision to be classified in a list. 11 days after the mainshock, on 14 April 1949 at 01h09 (Fig. S11) and 05h12 (Fig. S12), the earth shook again in Havré with magnitudes of $M_W$=3.5 and $M_W$=3.6. The macroseismic data coverage for these events is poor, but still a maximum intensity of respectively V and VI has been reported.





### 4.7 The October 1952 earthquake sequence in the Borinage

In October 1952, three earthquakes shook the Borinage area west of Mons. The first two occurred on 21 and 22 October ($M_W$=3.1 and $M_W$=2.8; Figs. S13 and S14), respectively at 21h15m and around 7h, and were moderately felt by the people. The third earthquake on 27 October 1952 at 6h11m ($M_W$=3.5; Fig. S15) was stronger and caused uproar among a part of the population who rushed out of the dwellings in the localities of Cuesmes, Flénu, Hornu, Jemappes, Quaregnon and Wasmes. The damage was limited to pieces of plaster falling from the ceilings, falling bricks, and falling pieces of chimneys in poor condition (intensity V).

### 4.8 The 10 July 1954 earthquake in the Borinage

On 10 July 1954 at 17h18m, another earthquake ($M_W$=3.5; Figs. 5 and S16) shook the same area than the 1952 events, with consequences relatively similar to those observed during the 27 October 1952 event. The local authorities paid much attention to properly filling the ROB official survey and indicated precise numbers on the damage to chimneys, indicating a slightly larger damage. We estimated intensity to V-VI in the localities of Quaregnon and Ghlin where the earthquake damaged 25 and 7 chimneys, respectively.

### 4.9 The 15 December 1965 earthquake near Strépy-Bracquegnies

On 15 December 1965 at 12h07m, a violent $M_W$=4.0 earthquake (Figs. 6 and S17) that lasted several seconds shook the region west of La Louvière and caused considerable commotion throughout the region (*"L'Indépendance (Edition du Centre) - 16/12/1965"*). There was quite some damage, especially to chimneys and roofs, but also to verandas damaged by falling chimneys. Few casualties occurred as people were hit by pieces of glass from shattered windows or skylights. The damage was the most important in Strépy-Bracquegnies. In this locality, there were overturned chimneys in practically every street, cracks in several buildings and many broken windows. Fallen stones and bricks damaged several cars. The ROB official questionnaire mentions 230 damaged and 122 overturned chimneys, which corresponds to 10% of the dwellings in the locality. The reported percentage is similar in the neighbouring commune of Bray. For these two localities, we assessed intensity as VII in EMS-98. We evaluated intensity as VI in Maurage and Trivières where the earthquake caused deep cracks in bricks and concrete walls in some houses, and damaged or overturned chimneys in 2 to 3%, respectively, of the total number of habitations. In Trivières, vials were falling off the shelves in a pharmacy, while someone had to hold the bottles of wine that were falling from the shelves in a store. Minor damage was observed in surrounding localities of Binche, Boussoit, Estinnes-au-Mont, Haine-Saint-Paul, Haine-Saint-Pierre, Houdeng-Aimeries, Houdeng-Goegnies, La Louvière, Leval-Trahegnies, Le Roeulx, Mont-sainte-Aldegonde, Morlanwelz-Mariemont, Péronnes-lez-Binche, Ressaix, Thieu, Vellereille-les-Brayeux, Villers-Saint-Ghislain and Waudrez.

Miners working in the region's collieries also perceived the earthquake. This was the case at the Quesnoy collieries in Trivières and at floors 872 and 1025 of the St-Marguerite coal mine in Péronnes-lez-Binche (see inlet in Fig. S17). The farthest locations from the epicentre where the ROB retrieved mentions of the earthquake were Wauthier-Braine (29 km) to the north,

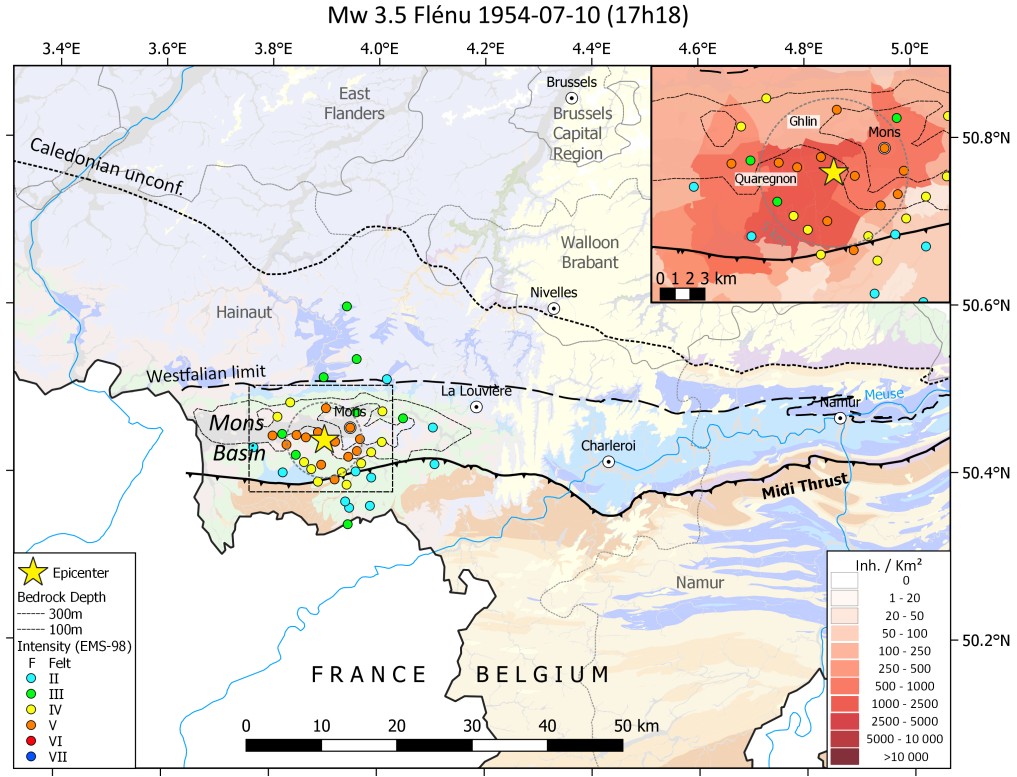

**Figure 5.** Macroseismic map of the 1954 $M_W$=3.5 earthquake in the Borinage (nr 16 in Table 1). Maximal intensity = V. Geology in background based upon http://www.onegeology.org/. Reproduced with the permission of OneGeology. All rights Reserved. The inlet shows the population density.

Viesville (21 km) to the east, Forge-Philippe (55 km) to the south, and Grandmetz (40 km) to the north-west. Remarkably, also the city of Ghent replied to the survey (73 km). The earthquake was followed the same day by three felt aftershocks.

### 4.10 The 16 January 1966 earthquakes in the La Louvière-Centre Basin

One month after the earthquakes in Strépy-Bracquegnies, the earth shook the region a few km more to the East. On 16 January 1966, two earthquakes occurred in the morning, the first one with $M_W$=2.9 at 0h13m (1h13m local time; Fig. S18) and the second one of $M_W$=3.5 at 6h51m (7h51m local time; Fig. S19) near Morlanwelz-Mariemont. The first seismic event woke up part of the population in La Louvière and nearby localities. The second earthquake was stronger and caused damage to few chimneys and the falling plaster inside few houses in the locality of Haine-Saint-Pierre (intensity V), while it was largely felt

in the localities of La Louvière, Chapelle-lez-Herlaimont, La Hestre, Jolimont, Haine-Saint-Paul and Manage.

These two events preceded a $M_W$=4.0 earthquake (Fig. S20) that occurred at 12h32m and caused a great deal of emotion and, in some places, even panic among the population. Indeed, in addition to minor incidents, such as falling frames, untimely


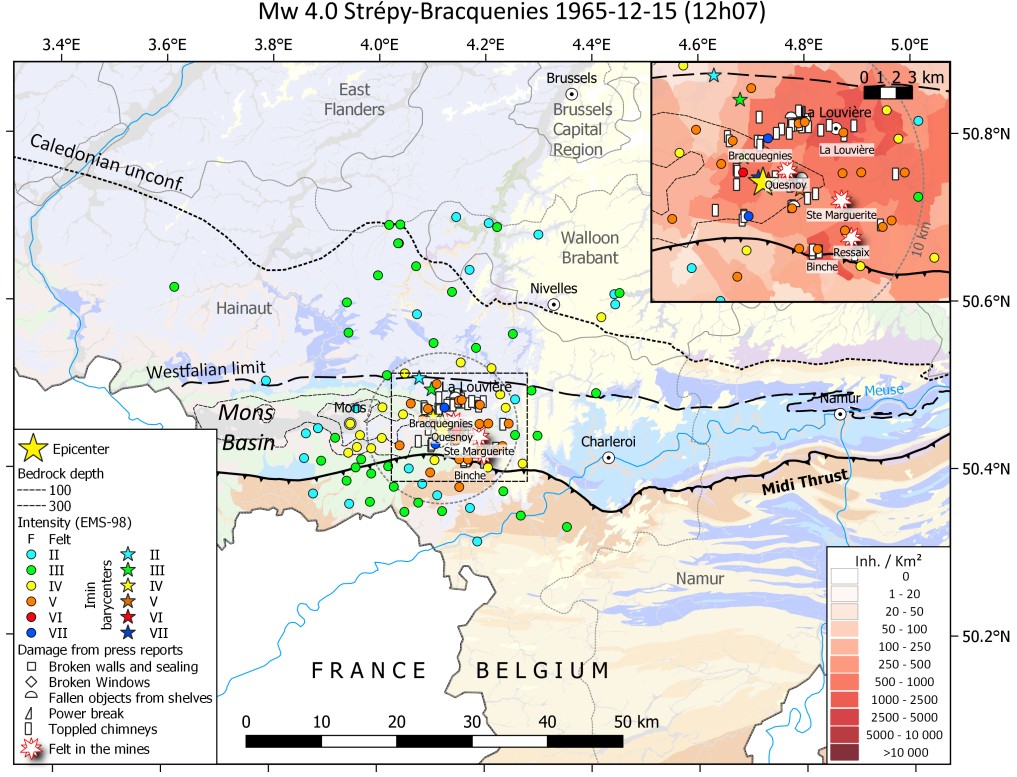

**Figure 6.** Macroseismic map of the 1965 $M_W$=4.0 ($M_L$=4.4) Strépy-Bracquegnies earthquake (nr 17 in Table 1). Maximal intensity = VII. Geology in background based upon http://www.onegeology.org/. Reproduced with the permission of OneGeology. All rights Reserved. The inlet shows localities where damage has been reported in press reports with population density as background.

clattering of glasses, vases and broken dishes, there were other more serious accidents. In Chapelle-lez-Herlaimont, Carnières, Morlanwelz, the material damage is quite considerable, although not very spectacular but, fortunately, there were no accidents

to persons. Throughout the affected region, the tremor also caused a power failure and electricity only restored after ten minutes to an hour (*"Le Rappel" - 17/01/1966*). The official ROB inquiry indicates that the shock damaged or overturned more or less 400 chimneys in Carnières, which corresponds to 14% of their total number in the locality. In Morlanwelz-Mariemont this percentage is smaller, around 7-8%. We assessed intensity as VII in these two localities. In Chapelle-lez-Herlaimont and Bellecourt, we evaluated intensity as VI based on the percentage of damaged and overturned chimneys, which is nearly 3%.

Minor damages are reported in La Louvière, Haine-Saint-Pierre, La Hestre, Fayt-Lez-Manage, Manage, Piéton, Souvret and Trazegnies (intensity V).

The earthquake was felt farther to the north (up to 52 km), than to the south (up to 17 km). The northwards shift of the intensity II barycentre with respect to the epicentre shows that this event was farther felt in the borders of the Brabant Massif than in the coal mining area east and west or in the Ardennes to the south.





## 4.11 The 28 March 1967 15h49m earthquake in the La Louvière-Centre Basin

This $M_W$=4.1 earthquake (Figs. 7 and S21) is the strongest earthquake that occurred in the Hainaut coal area, with a similar magnitude than the 1949 Havré earthquake. The shaking was of particular violence in the region between La Louvière and Charleroi. The paper *"La Nouvelle gazette"* of 29/03/1967 reports: *"... the earthquake lasted about ten seconds, which was very frightening for a large part of the population of this region; the ground was in fact tilting underfoot and inside the buildings it seemed that the walls would not be able to resist the telluric movement. In some places, the power was cut off abruptly. Frightened, some inhabitants rushed out of their homes, while others sought refuge in their cellars."*

The most affected localities are Carnières, Morlanwelz-Mariemont and Trazegnies where the percentage of damaged or completely destroyed chimneys range from 8 to 10% (intensity VII). Inside many houses, ceilings and walls were also cracked. Fortunately, there were no personal accidents. However, emotion was very strong everywhere. In Fontaine-l'Evêque, Piéton, Chapelle-lez-Herlaimont, La Hestre and Godarville where damage is smaller, we evaluated intensity to VI.

The farthest locations from the epicentre where we retrieve mentions of the earthquake are Forest (39 km) to the north, Bonneville (54 km) to the east, Gozée (15 km) to the south, and Ville-sur-Haine (16 km) to the west. The barycentres of intensities IV, III and II are shifted northwards, showing the low attenuation properties of the Brabant Massif (Fig. 7). The earthquake was followed by many aftershocks recorded at the seismic station of Dourbes (Camelbeeck, 1985; 1993). One of these events of magnitude $M_L$=3.3 occurred on April 4 at 18h04. It was felt by the inhabitants of La Louvière, Carnieres and Morlanwelz but did not cause any damage.

## 4.12 The August and September 1968 earthquakes near La Louvière

Another series of (damaging) earthquakes occurred in the summer and fall of 1968 near La Louvière. The sequence began at 7h26m on 12 August 1968 (Fig. S22) with a $M_W$=3.6 earthquake causing panic of people that rushed on the threshold of the houses. The ROB official survey mentions a few damaged chimneys in La Louvière, Haine-Saint-Pierre, Haine-Saint-Paul and Morlanwelz-Mariemont, which suggests that intensity could have reached V in these localities.

On August 13, a first quite violent shock of $M_W$=3.6 occurred at 16h17m and shook the locality of La Louvière, with a slight extension in Haine-Saint-Pierre and Haine-Saint-Paul. The rumble was brief and spectacular manifestations were limited. This event was followed by a lighter, barely perceptible tremor of $M_W$=3.0 at 16h40m.

The $M_W$=3.9 earthquake of 16h57m (Fig. S23) was stronger and damaging. In La Louvière, there was a brief moment of panic at the time of the tremor. The facades of houses vibrated, chimneys collapsed and in the workbenches of some shops, there was upheaval. The localities of Haine-Saint-Pierre and Haine-Saint-Paul also suffered from the earthquake. Chimneys fell down everywhere, but fortunately, there were no injuries. A consequence was also that all the telephone switchboards of the fire departments of La Louvière and Morlanwelz were overwhelmed (*"L'Indépendance (Edition du Centre) - 14-15/08/1968"*). Even if the number of damaged and overturned chimneys exceeded a few hundred in La Louvière, Haine-Saint-Pierre and Morlanwelz-Mariemont, this damage concerns the most inhabited part of the area and it only corresponds to a few percent of the dwellings. Hence, we estimated intensity as VI in those localities.

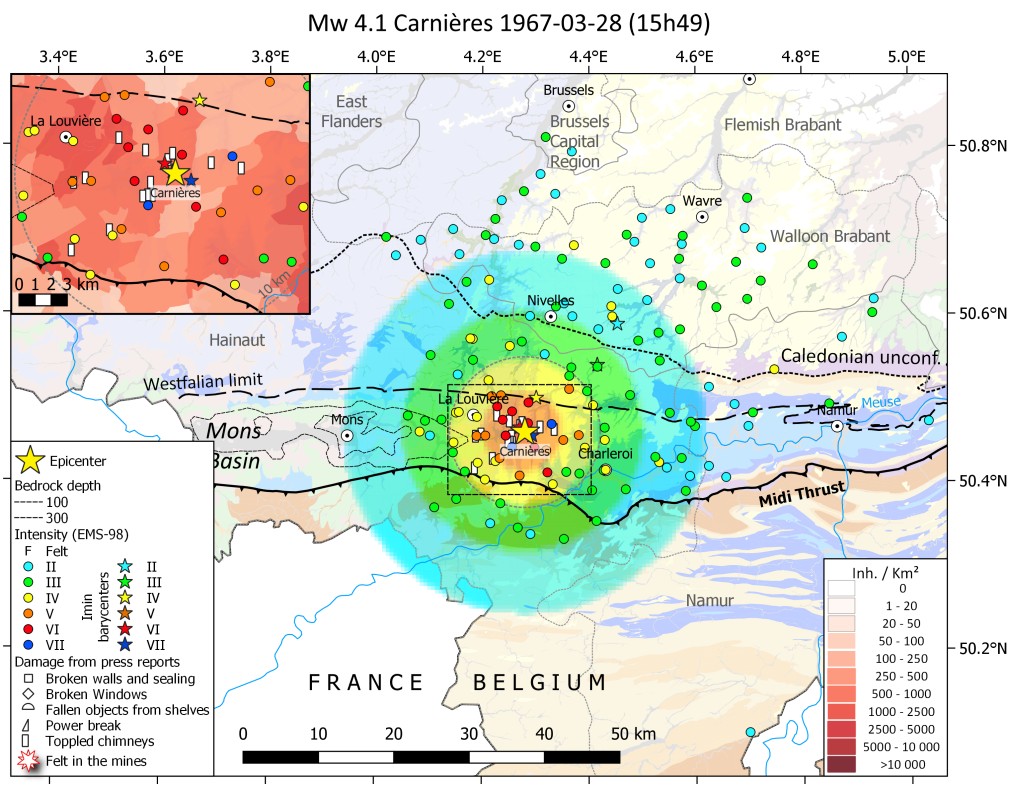

**Figure 7.** Macroseismic map of the 1967 $M_W$=4.1 ($M_L$=4.5) Carnières earthquake (nr 21 in Table 1). Note the asymmetric macroseismic field: this event has been felt more northwards within the borders of the Brabant Massif, than southwards in the Ardennes, which results in a northwards shift of the lower intensity (Imin IV, III and II) barycentres. The inlet shows localities where damage has been reported in press reports. In the background, the Hainaut intensity attenuation model developed in this study (see section 7.2) is applied to the parameters of this event. Note that this attenuation model only can be applied within the coal area (between the Midi Thrust and Westfalian limit). Modelled Imax = VI, but locally intensity VII is observed. Geology in background based upon http://www.onegeology.org/.Reproduced with the permission of OneGeology. All rights Reserved.
During the next period of approximately two months, the seismic station in Dourbes recorded a series of earthquakes. Two of them were felt in the La Louvière area and occurred on 23 September at 4h08m ($M_W$=3.2; Fig. S24) and 5h47m ($M_W$=3.0: Fig. S25).

### 4.13   The 3 November 1970 earthquake near Charleroi

This $M_W$=3.6 earthquake (Fig. S26) was strongly felt and caused slight damage in the cities of Dampremy, Marchienne-au-Pont, Marcinelle, Monceau-sur-Sambre and Mont-sur-Marchienne located south of Charleroi. Many people left their homes. Damage is limited to cracks in plastered walls, small plaster fragments falling from the ceilings, cracked or broken windows, bricks falling, or few falling chimneys that were in bad condition. We assessed intensity as V in those localities.

### 4.14   The 24 October 1976 earthquake south of the coal area

This $M_W$=3.9 earthquake (Fig. S27) occurred a few km south of the coal area. It strongly shook villages near the Belgian-France border. No damage was reported. The BCSF [Bureau Central Sismologique Français] conducted an inquiry on the effects of this event in France (BCSF, 1983). We used their intensity evaluations to extend macroseismic information on the French territory.

### 4.15   The 14 September 1982, 4 and 9 August 1983 earthquakes south of the coal area

The last earthquake in the Hainaut coal area for which it was possible to provide a macroseismic map occurred near Carnières on 14 September 1982 at 19h24 ($M_W$=3.4; Fig. S28). Two earthquakes were also widely felt in the region of Charleroi on 4 and 9 August 1983, but only few testimonies on their effects and very few positive answers to the ROB surveys were collected.

## 5   Intensity attenuation and focal depth estimation

Seismic intensity is an empirical measure of the severity of ground motions generated by earthquakes. Determining intensity inside the radius of an earthquake's perceptibility allows mapping ground motion strength and its spatial variability. The macroseismic field directly relates to earthquake epicentre location, focal depth and magnitude, and near-field energy absorption coefficient (Ambraseys, 1985). Hence, determining the parameters controlling seismic energy absorption offers the perspective to evaluate the location and magnitude of past earthquakes from their intensity spatial distribution. It also gives the possibility to predict intensities for specific earthquake models with given focal depth and magnitude.

### 5.1   Methodology

Ambraseys (1985), Hinzen and Oemisch (2001), Bakun and Scotti (2006), and Stromeyer and Grünthal (2009) developed regional intensity attenuation models using earthquake datasets from Western and Central Europe. Except for Ambraseys (1985), who used isoseismal radii, these authors all based their models on IDP distributions. Knuts et al. (2016) and Camelbeeck





et al. (2021) successfully applied these models to determine epicentral locations and magnitudes of historical earthquakes in Belgium.

Even though the datasets used to compute these models also include information on very shallow earthquakes, the small number of shallow events with respect to deep ones, makes these models less suitable to simulate the macroseismic field of
shallow earthquakes. However, seismic attenuation characteristics are more variable in the fractured upper layers of the crust because of large lateral variations of mechanical characteristics of rocks and sediments near the surface. Hence, for shallow earthquakes, it would be more appropriate to develop a new local intensity attenuation model than using these Western and Central Europe models. Moreover, given the large available intensity dataset for the Hainaut coal area, it would be even more realistic (Table 2).

To develop a local Hainaut intensity attenuation model, we used the classical formulation developed by Kövesligethy (1907) and still widely used today (e.g. Ambraseys, 1985; Stromeyer and Grünthal, 2009):

$$I = I_0 - a * log(\sqrt{\frac{R^2 + Z^2}{Z^2}}) - b * (\sqrt{R^2 + Z^2} - Z) \tag{1}$$

where $I$ is the intensity at epicentral distance $R$ from an earthquake source at a focal depth $Z$, $I_0$ is the epicentral intensity strength, $a$ and $b$ are parameters that respectively correspond to the multiplication of the geometric spreading and energy
absorption factors by the proportionality factor between intensity and ground acceleration (Ambraseys, 1985; Stromeyer and Grünthal, 2009). These parameters $a$ and $b$ can be derived by fitting Eq. 1 to IDPs of calibration earthquakes with a well-determined location and focal depth. Solving the parameters of Eq. 1 using intensity datasets can performed by three different approaches: (1) using intensities and epicentral distances of all individual observations; (2) using the mean distance and its standard deviation by intensity binning; and (3) using the mean intensity and its standard deviation by distance binning (used
in this work).

## 5.2 Intensity attenuation in the Hainaut coal area

Here, we represent our dataset by applying IDP epicentral distance binning. Figure 8 presents an example for the 15 December 1965 earthquake (macroseismic map on Fig. 7). For each distance bin of 2.5 km, the diagram reports the mean intensity minus $I_0$ (determined from the IDP distribution - see further in this section) and its standard deviation, which mostly represents the
intensity variability inside the distance bins. The number of IDPs in each bin progressively increases up to a distance of 15 km from the epicentre and then abruptly decreases. Beyond this distance, there are only a few IDPs, which are of low intensities, which indicates that the earthquake was likely not felt in many localities contributing to these bins. This suggests that the mean values computed from these IDPs would overestimate the current mean intensity of the bins because "not felt" localities are not included in the computation. This example also shows the rapid decrease of intensity with distance in the coal area (Fig. 8),
which for the 15 December 1965 earthquake corresponds to a decrease of three intensity grades on a distance range of 15 km. To the north of the Hainaut coal area, inside the borders of the Brabant Massif (see e.g. Figs. 4, 6 and 7), the largest earthquakes are weakly felt with intensity II to III up to distance exceeding 40-50 km, suggesting that intensity attenuation is lower than in

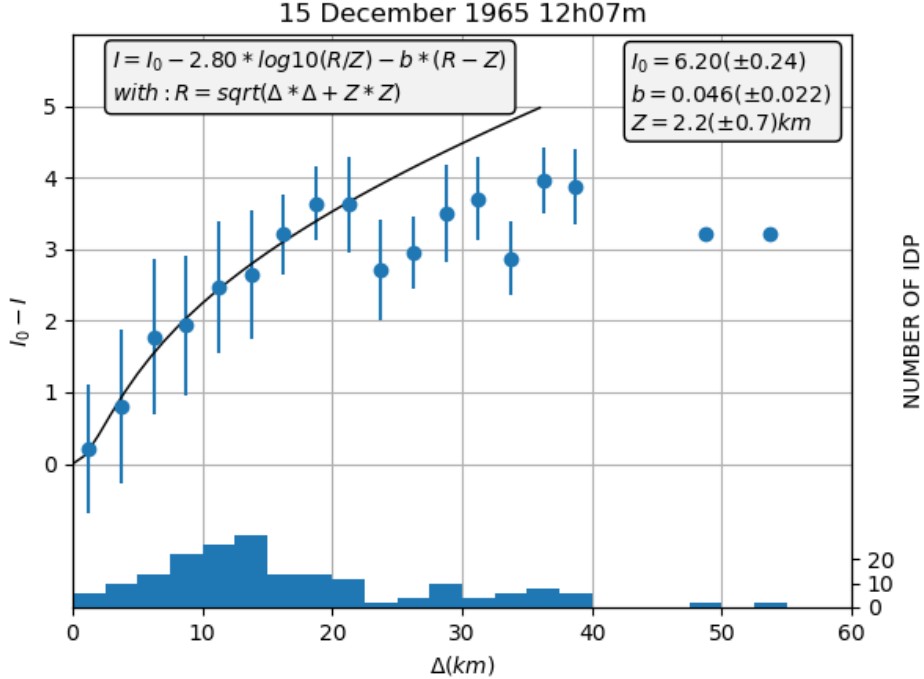

**Figure 8.** Intensity attenuation of the 15 December 1965 Strépy-Bracquegnies earthquake expressed as mean intensity change relative to $I_0$ (blue dots) calculated for bins of 2.5 km (histogram). Vertical blue bars show intensity standard deviation for each distance bin that expresses the intensity variability in the bin. The legend reports local parameters fitting the intensity attenuation of Eq. 1 with $a$ fixed to 2.80.

the coal area. South of the Hainaut coal area, the Midi fault (Fig. 2) seems to play the role of a seismic barrier and intensity decays more rapidly in the Ardenne Massif than in the Brabant Massif. This observation was already done by (Charlier, 1951).

Then, there are two reasons of not using IDPs at distances larger than 15 km for an intensity attenuation analysis in the coal area: (1) IDPs beyond these distances bias the mean intensity values in the bins, and (2) the intensity attenuation of the coal area differs from attenuation in the Brabant Massif and in the Ardenne. Hence, applying a distance range larger than 15 km would not properly model the attenuation in the coal area, but would provide an intermediate attenuation including crustal characteristics from these three areas. Moreover, the distance binning of intensity with a short distance range of 2.5 to 3 km is

more appropriate in our case to invert the parameters $a$ and $b$ of Eq. 1 than an intensity binning because it will furnish more data points. In the case of the 15 December 1965 earthquake (Fig. 8), 6 bins allow a better fitting with Eq. 1 than using only mean distance for the 4 intensity bins covering 3 ranges of intensity values.

  The dataset used for this modelling approach is relatively small, with 76 mean intensity change values obtained by distance binning of 12 earthquakes. In our computation we also included two additional events (identified by an ° symbol in Table 3)

because, although they occurred outside the Hainaut coal area, the geological context of the felt observations is similar to the





**Table 3.** *Depth evaluation of calibration earthquakes used for attenuation modelling. $I_0$: epicentral intensity; Z: depth; b: b-value; step: length of distance bin; n: number of distance bins. (2) first step of the analysis; (2) second step of the analysis. °earthquake not included in the attenuation modelling but used for verifying the model.*

| id_earth | Date | Time | Lat (°N) | Lon (°E) | $I_0^{(1)}$ | Z (km)$^{(1)}$ | b (km)$^{(1)}$ | $I_0^{(2)}$ | Z (km)$^{(2)}$ | step | n |
|---|---|---|---|---|---|---|---|---|---|---|---|
| 465 | 1911-06-01 | 22h52m | 50.46 | 4.46 | 6.15±0.14 | 2.4±0.6 | 0.010±0.026 | 5.91±0.48 | 4.3±1.8 | 2.5 | 5 |
| 534 | 1949-04-03 | 12h33m | 50.45 | 4.07 | 7.24±0.37 | 1.7±0.8 | 0.064±0.034 | 6.95±0.52 | 2.2±0.8 | 2.5 | 7 |
| 549 | 1952-10-27 | 06h11m | 50.44 | 3.9 | 5.29±0.37 | 2.3±1.2 | 0.022±0.045 | 5.03±0.41 | 3.5±1.2 | 2.5 | 6 |
| 562 | 1954-07-10 | 17h18m | 50.46 | 3.88 | 5.43±0.37 | 2.3±1.1 | 0.060±0.032 | 5.39±0.44 | 3.3±1.2 | 3 | 5 |
| 582 | 1965-12-15 | 12h07m | 50.45 | 4.09 | 6.20±0.24 | 2.2±0.7 | 0.046±0.022 | 6.19±0.47 | 2.7±0.8 | 2.5 | 7 |
| 588 | 1966-01-16 | 06h51m | 50.46 | 4.23 | 4.86±0.19 | 2.1±0.6 | 0.023±0.030 | 4.77±0.56 | 3.3±1.6 | 2.5 | 5 |
| 589 | 1966-01-16 | 12h32m | 50.47 | 4.26 | 6.00±0.19 | 3.1±0.9 | 0.150±0.039 | 6.23±0.66 | 2.1±0.8 | 3.5 | 5 |
| 597 | 1967-03-28 | 15h49m | 50.45 | 4.27 | 6.68±0.79 | 1.8±1.7 | 0.112±0.073 | 6.21±0.47 | 3.0±1.0 | 3 | 7 |
| 603 | 1968-08-12 | 07h26m | 50.45 | 4.21 | 5.33±0.29 | 2.0±0.7 | 0.088±0.030 | 5.44±0.70 | 2.3±1.0 | 3 | 4 |
| 606 | 1968-08-13 | 16h57m | 50.46 | 4.23 | 5.81±0.27 | 4.0±1.9 | 0.162±0.067 | 6.01±0.54 | 2.3±0.8 | 2.5 | 6 |
| 607 | 1968-09-23 | 04h07m | 50.46 | 4.23 | 4.68±0.39 | 2.1±1.4 | 0.048±0.098 | 4.76±0.76 | 2.8±1.7 | 2.5 | 4 |
| 612 | 1970-11-03 | 08h45m | 50.4 | 4.41 | 5.16±0.35 | 3.2±2.1 | 0.089±0.093 | 5.29±0.63 | 2.3±1.0 | 2.5 | 5 |
| 627° | 1976-10-24* | 20h33m | 50.36 | 3.98 | 5.08±0.23 | 4.0±1.5 | 0.036±0.026 | 5.13±0.38 | 5.5±1.7 | 3 | 5 |
| 641° | 1983-11-08* | 00h49m | 50.63 | 5.51 | 7.13±0.18 | 3.3±0.9 | 0.044±0.017 | 6.91±0.28 | 5.7±1.5 | 3 | 7 |

earthquakes that occurred inside the Hainaut coal area. These two events are the 24 October 1976 earthquake (Fig. S27), which occurred a few km south of the Hainaut coal area, and the 8 November 1983 Liège earthquake (Camelbeeck, 1993; Camelbeeck et al., 2021, Fig. S29) that occurred in the Liège coal area area in a geological context similar to Hainaut.

The main hypothesis in our fitting analysis is that intensity attenuation is homogeneous in the Hainaut coal area, which
means that the parameters $a$ and $b$ have the same values for all seismic events in the area. Hence, the observed variations in the decay of intensity with distance between the different calibration earthquakes are only associated with a difference in focal depth, in the uncertainty of the attenuation model and in the data. We determine parameters $a$ and $b$ in two steps:

1. As focal depth is unknown for the calibration earthquakes, a first step in the analysis was to evaluate their depth by fitting each earthquake dataset to Eq. 1 (see macroseismic maps in the Atlas). As this equation has four unknowns
and the number of distance bins for each earthquake does not exceed seven, we fixed the value of the parameter $a$, and inverted the equation to evaluate the attenuation parameter $b$, the earthquake focal depth Z and epicentral intensity strength $I_0$. We considered that $b$ is more dependent on the highly variable material properties near the Earth's surface than $a$, which should be relatively similar in Hainaut than elsewhere in Europe. We adopted the value $a = 2.80$ of the WLQ model of Stromeyer and Grünthal (2009). Table 3 reports the results of this analysis. Our main conclusion is that
all the studied Hainaut earthquakes would have similar focal depth ranging between 1.6 and 4.0 km, with uncertainties around 1.5 km.





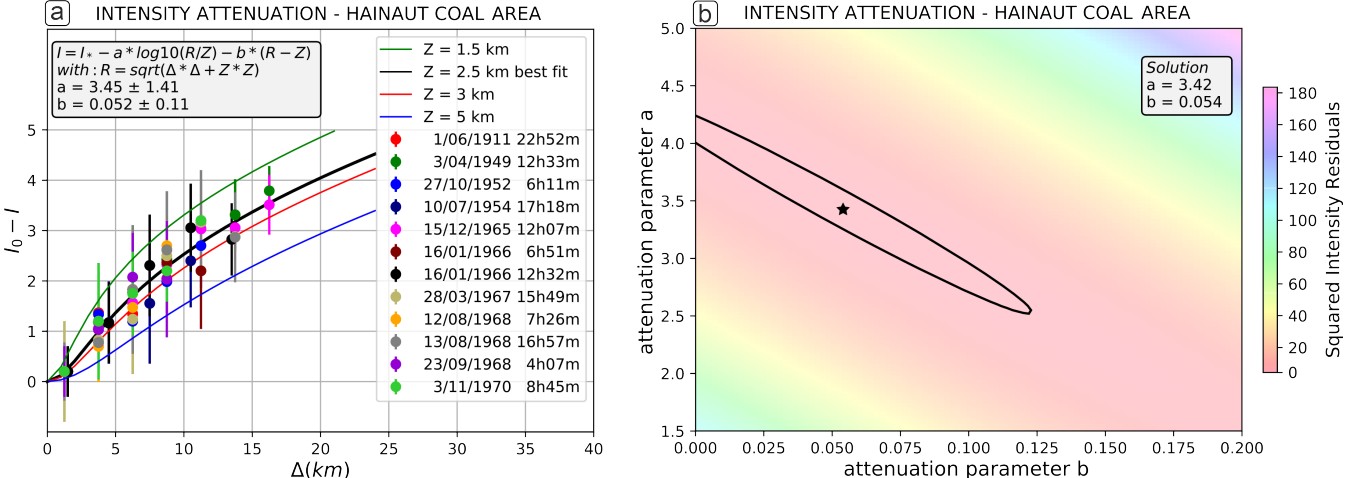

**Figure 9.** a) Fitting the intensity dataset of 12 calibration earthquakes to Eq. 1 to determine $a$ and $b$ attenuation parameters and the focal depth considering a uniform depth for all events. b) Least-squares fitting by sampling the $a$ and $b$ intensity parameters space: the solution is represented by the star and the black ellipse limits the 0.95 confidence region.

2.  In the second step, we considered that the 12 calibration Hainaut earthquakes have the same focal depth, which is supported by the results of the first step of the analysis. We fixed the value of $I_0$ by considering that the mean intensity of the first distance bin of each earthquake is equal to $I_0 - 0.3$ based on the results of the first step. We represent this estimation of $I_0$ by $I^*$. Then, we invert the complete dataset to evaluate $a$, $b$ and the focal depth, identical for all the earthquakes, that minimised the residuals by a Least-squares modelling. Figure 9a presents the results of the inversion: $a$=3.45±1.41 and $b$=0.052±0.11, while the focal depth that best fits the data is 2.5 km. The relative small number of data and the lack of information at distances larger than 20 km cause the large uncertainties on $a$ and $b$. However, it relies on their relative dependence which is well illustrated by their joint confidence region in Figure 9b. Figure 10 presents the intensity attenuation curves corresponding to the best solution and the two extreme solutions at the 0.95 confidence region for focal depths ranging from 1 to 6 km. The difference between these models are very small for distances less than 15 km, i.e. 0.3 intensity units for a distance of 20 km, but becomes more important at larger distances. The uncertainty on the two parameters reflects the fact that $a$ controls the short distance behaviour and is better determined while $b$ characterises the curves at large distance.

## 5.3 Earthquake focal depth

Figure 10 reports the influence of focal depth from 1.0 to 6.0 km on the intensity attenuation curves and its stronger effect on the attenuation function than the uncertainties on the attenuation parameters. This also suggests that earthquake focal depth can be evaluated with a good accuracy using IDPs and that the differences in attenuation observed between the different earthquakes in the modelling (Fig. 9a) would likely reflect the small differences in their respective focal depths. Subsequently, we used the

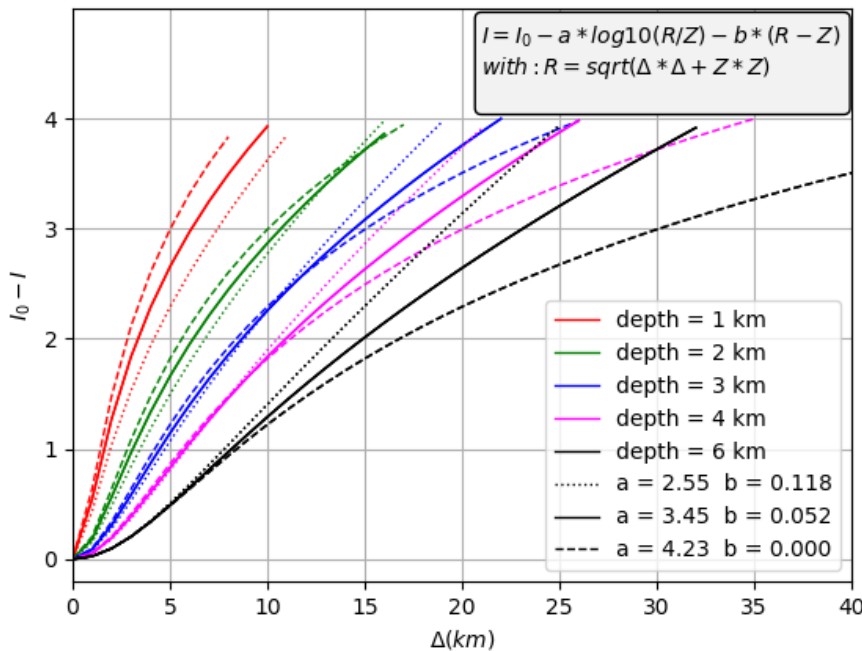

**Figure 10.** Variation of $I_0$ - $I$ in function of epicentral distance corresponding to intensity attenuation models of Figure 9. The curves correspond to the best fitting solution (full lines) and the two extreme solutions (dotted and dashed lines) at the 0.95 confidence region for focal depths ranging from 1 to 6 km.

new Hainaut attenuation model to estimate the focal depth and the epicentral intensity strength of the 12 reference earthquakes in the Hainaut coal area and the 1976 and 1983 Liège earthquakes. Figure 10 presents the results of this modelling for the 15 December 1965 earthquake. In the Atlas, the same diagram is provided for the 13 other earthquakes. For all the other events in Table 1 than the 12 calibration events, macroseismic datasets are less complete and the full modelling cannot be applied. Nevertheless, the available information is sufficient to correctly evaluate focal depth for most of them (see focal depth in

brackets in Table 1). For each event, the input data for focal depth determination are Imax, the maximal observed intensity, and the intensity $I$ and epicentral distance $\Delta$ for each observed IDP. Then, for each of these events, we created 250 different datasets by adding a random noise with possible values of -0.5, 0 or +0.5 to the intensity $I$ of the IDPs, which would represent the uncertainty on each intensity evaluation. For each of those modelled IDPs based on the current ones, we searched the focal depth Z minimizing $I - I_{Max} = -3.45 * log(\Delta/Z) - 0.052 * (\Delta - Z)$ by testing focal depths by a range of 0.1 km from 0 to 10

km. The computed mean focal depths and the sigma value of the distribution from the 250 different models for each earthquake are indicated in Table 1 inside brackets. Some other earthquakes, like the events in Havré and Fleurus in 1887 and 1904, or events that occurred between 1950 and 1960, sometimes slight damage was reported. From the estimations of Imax and the



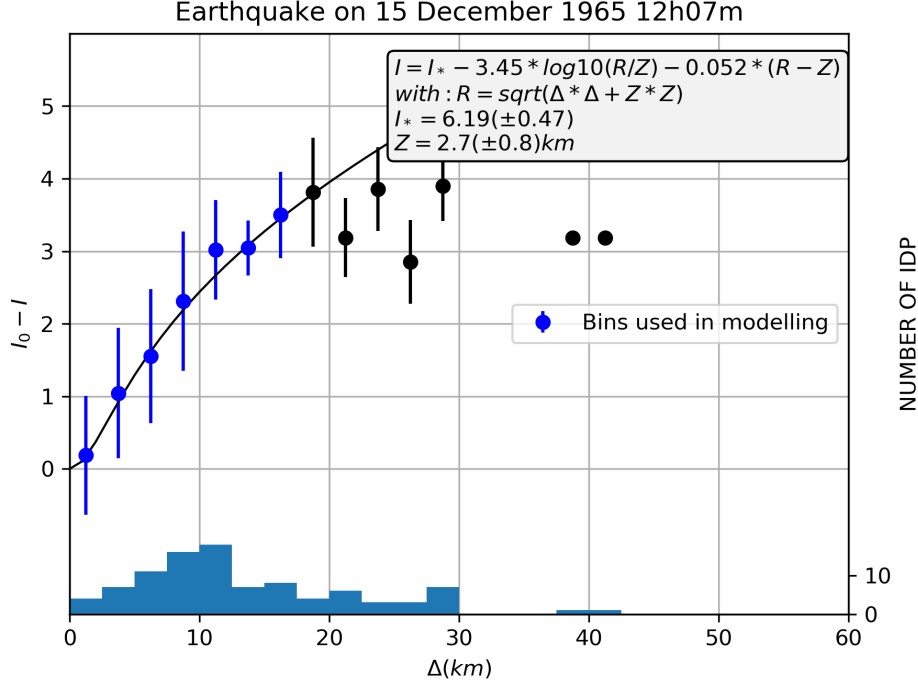

**Figure 11.** Evaluation of focal depth and epicentral intensity strength for the 15 December 1965 earthquake. The first seven distance bins are used in the modelling. Similar diagrams are provided in the Atlas for the 13 other earthquakes for which this method was used.

published perceptibility radius, we also evaluated their focal depth, that are very shallow. For these events, we only indicate the estimated focal depth inside brackets in Table 1 and in the full catalogue in the Supplement, but without any uncertainty.

## 6 Instrumental magnitudes and magnitude determined from macroseismic data


Camelbeeck (1985a, 1993) determined the local magnitude $M_L$ of the Hainaut earthquakes between 1911 and 1985 when the seismic measurements from at least one seismic station were available. For some events, it was also possible to determine surface wave magnitude MS using the formula of Kárník (1971). Moreover, Camelbeeck (1985b) estimated seismic moments for 17 earthquakes that occurred between 1965 and 1970 in the Hainaut coal area based on the coda waves enveloppe measured on the paper recordings from the Belgian seismic station of Dourbes. Even if the absolute value of these seismic moments were dependent on approximate parameterisation of the scattering properties of the crust between the coal area and the town of Dourbes, the used method furnishes a reliable ratio of the seismic moment values between the different earthquakes. Denieul (2014) used the recordings of the CEA-LDG (Commissariat à l'Energie Atomique, Laboratoire de Détection et de Géophysique, France) seismic network to determine moment magnitudes of significant earthquakes in France and surround-




ing regions that occurred from 1963 to 2013. This study determined the moment magnitude $M_W$ for the three earthquakes in Hainaut that occurred on 15 December 1965 at 12h07m, 16 January 1966 at 12h32m and 28 March 1967 at 15h49m as respectively 4.0, 4.0 and 4.1 with a one sigma uncertainty of 0.2. These results suggest that the moment magnitude determined from Camelbeeck (1985b) should be diminished by a constant factor of 0.3 magnitude units. This result also allows reevaluating the relationship between $M_L$ and $M_W$ for the Hainaut earthquakes furnished by Camelbeeck (1985b) as:

$$M_W = 1.294(\pm0.08) + 0.610(\pm0.059) * M_L \tag{2}$$

which is valid between $M_L$=2.6 to $M_L$=4.6.

We reported in Table 1 the instrumental magnitude values that were determined for earthquakes in the Hainaut coal area. In addition, we used Eq. 2 to estimate $M_W$ for the earthquakes for which only $M_L$ was determined. For those events, the $M_W$ value and its uncertainty are indicated inside brackets, while $M_W$ determined from Camelbeeck (1985b) modified by Denieul

(2014) are reported with their uncertainty without brackets.

Thanks to the fact that instrumental magnitudes were determined for a part of the earthquakes for which macroseismic data are available, we were able to establish relationships between earthquake magnitude and macroseismic parameters and then to determine a formula for a robust evaluation of earthquake magnitude $M_W$ directly from macroseismic information for events that were not recorded by seismic stations.

We used the classical model (Sponheuer, 1962; Van Gils and Zaczek, 1978; Ambraseys, 1985; Stromeyer et al., 2004):

$$M = a * I_0 + b * log(h) + c \tag{3}$$

which determines the magnitude knowing epicentral intensity strength $I_0$ and focal depth $h$. As the range of focal depth in our calibration dataset of 12 earthquakes is limited around 2.5 km (Figure 10), it was not possible to find a reliable relationship with focal depth. Then, we preferred first to not consider it in the modelling. However, $I_0$ is a parameter resulting from the

fitting of IDPs with Eq. 1 (1) and hence cannot be determined for earthquakes with only few IDPs (e.g. for $19^{th}$ or first half of $20^{th}$ century earthquakes or aftershocks of strong earthquakes). In this case, the only available parameter is the maximal observed intensity Imax (see Table 1). For this reason, we established a relationship between $M_W$ and Imax (Fig. 12) rather than $I_0$ so that a specific model can be used for earthquakes with few macroseismic observations:

$$M_W = 1.744(\pm0.130) + 0.346(\pm0.098) * I_{max} \tag{4}$$

This relationship is certainly valid for earthquakes with focal depths in the range 1.5 to 4.0 km as the ones in our calibration dataset and their associated seismic sequences, but it would overestimate the magnitude for earthquakes closer to the surface. Considering that geometrical spreading would play a more significant role in seismic waves energy attenuation from the earthquake depth to the epicentre at the surface and that body waves is a major part of the radiated energy to the surface, $b$ is fixed to 2.0 in Eq. 3.

Hence, for earthquakes shallower than 1.5 km, we determined $M_W$ using the relationship:

$$M_W = 0.948(\pm0.130) + 2.0 * log(h) + 0.346(\pm0.098) * I_{max} \tag{5}$$


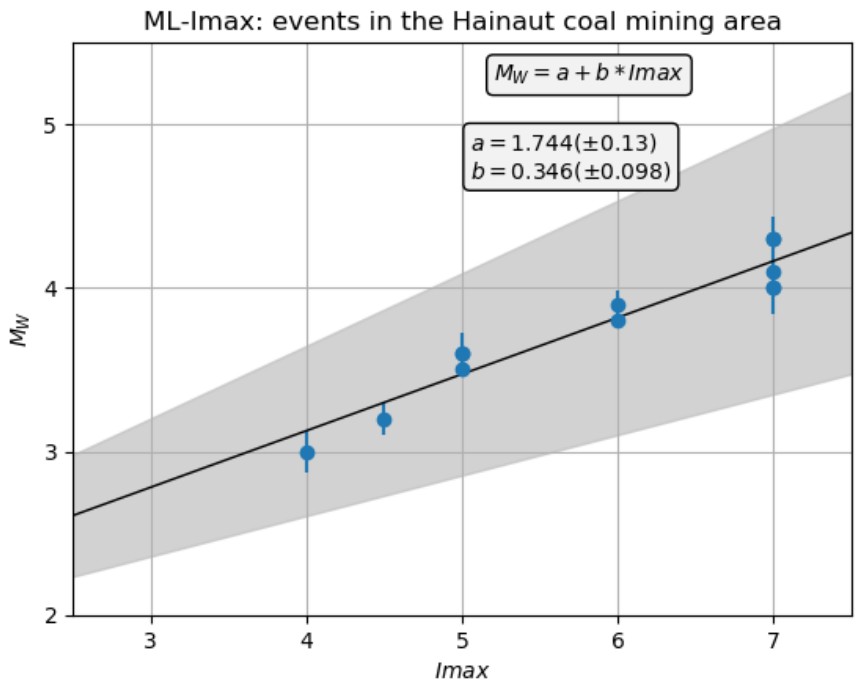

**Figure 12.** Relationship between $M_W$ and Imax (approximation for $I_0$) determined for 12 calibration earthquakes in the Hainaut coal area (three data points are not visible because they are superposed above each other).

In Table 1, all the earthquakes for which $M_W$ was determined using macroseismic information and Eq. 4 or Eq. 5 are reported in the column $M_W$_m.

## 7 Discussion

In this discussion, we emphasise three aspects of the seismicity that occurred in the Hainaut coal area between the end of the $19^{th}$ century and 1983. First, we analyse the cumulative impact of this Hainaut coal area seismicity and compare it to the effects of a few larger magnitude $20^{th}$ century earthquakes that occurred elsewhere in Belgium, suggesting that this seismicity could be overestimated in current seismic hazard maps. Second, we discuss the pertinence of our new Hainaut intensity attenuation law at the light of the spatial resolution of our intensity dataset and the validity of this intensity attenuation law outside the coal
area. Last, we underline the importance of our focal depth determinations to discuss the causes of the seismicity in and near the Hainaut coal area.



## 7.1 The impact of the Hainaut seismic activity

There is no doubt that the shallowness of the seismic events in Hainaut is the dominant factor that explains the locally observed severity of damage to buildings for events with such relatively small magnitudes. The damaging character of this seismicity is

well illustrated on Figure 13 which shows the maximum intensity observed within each commune in the Hainaut coal area for all 123 events of the Hainaut seismic catalogue. Maximum intensity equal or greater than V was observed in all the localities in a 60 km long and 15-20 km wide range of the coal area, which extends from 10 km east of the French border to 15 km west of the city of Charleroi. Outside the coal area, this seismicity had no damaging impact and in only a few communes, intensity V was observed. The area between Mons and Charleroi and centred on La Louvière was the most affected part with

a widespread repartition of maximal intensity VI, including some localities where intensity VII was observed. In the Borinage basin, intensity VI was only observed locally in a few communes in its western part. The most destructive events occurred during or at the end of the mining exploitation. This explains why they were rightly or wrongly associated with this industry and was at the origin of many complaints by the population against this industry. Moreover, as these events often occurred in seismic sequences that sometimes lasted several weeks, it aggravated the way this seismicity was experienced by people:

repetition of shakings, waking up during the night, and also the increasing damage that sometimes led to the ruinage of some houses. This was particularly true during the Havré sequence in 1949.

    Mining caused a lot of other environmental problems with consequences that were more dramatic than the shallow earthquake activity. Many buildings in the Hainaut coal area were damaged due to underground progression of the coal exploitation and the progressive settling that follows. Troch (2018a, b) presents the example of the locality of Gosselies, which was com-

pletely devastated between the two world wars because of the extensive coal production. In some areas, mine subsidence led to surfacing groundwater and increased the risk of flooding. It was necessary to evacuate the water by pumping systems otherwise wetlands, marshes, swamps, ponds and lakes appeared in the affected area (Troch, 2016). The subsidence and the permanency of humidity in some areas caused by mining activities are factors affecting the resistance of buildings, particularly to earthquakes, which partly explains the importance of some local damage reported after some earthquakes.

Apart from the Hainaut seismicity, also two other earthquakes had a strong local impact on the Hainaut coal area during the $20^{th}$ century: the strongly damaging 11 June 1938 Zulzeke-Nukerke $M_W$=5.0 (S30) and the 20 June 1995 $M_W$=4.1 Le Roeulx (S31) earthquakes in the Brabant Massif (Fig. 1). Both events took place much deeper ( 20 km and 25 km, respectively) and had a totally different effect than the Hainaut earthquakes because they were felt in large areas, yet they caused local damage in the coal area as shown for the 1938 earthquake on Figure 13b. This earthquake occurred some 40-45 km north-west of the

western extremity of the Hainaut coal area. It was reported in the coal area with intensities V in many localities, and intensity VI observed in some of them, mostly in its western part. In the Borinage, the slight damage caused by the 1938 earthquake is equivalent or even larger than the maximal impact of the Hainaut seismicity. Outside the coal area, the impact of the 1938 earthquake is larger everywhere. Similar conclusions arise from the few original documents concerning the effects of historical earthquakes. Some of them had a larger impact in the coal area than individual earthquakes observed in the Hainaut coal area

since the end of the $19^{th}$ century. Apart from the effects of the 23 February 1828 earthquake (see section 3.1), the earthquake





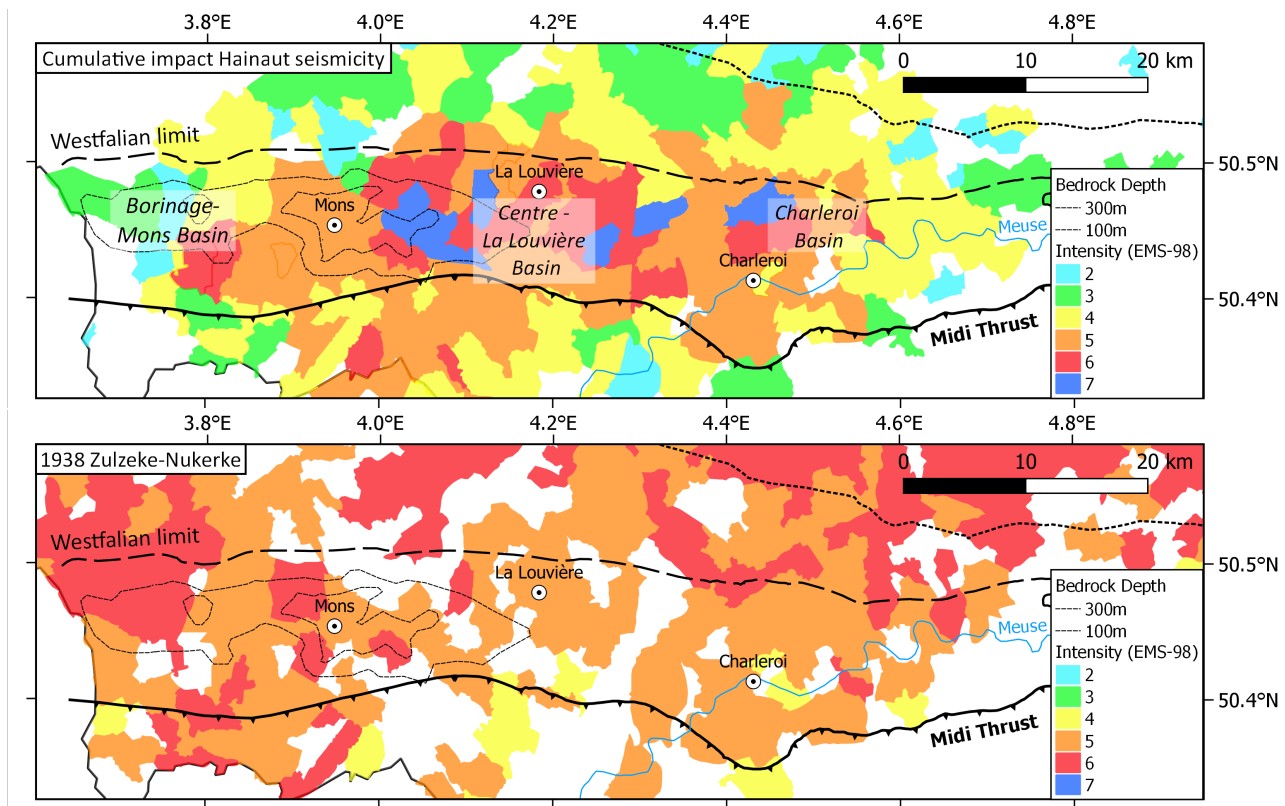

**Figure 13.** Top) Communal map of the Hainaut coal area showing the maximum intensities that were reached by the 123 earthquakes. Bottom) for comparison, the impact of the 1938 earthquake on the Hainaut coal area is shown. This earthquake had a larger impact on the Brabant Massif and in the western part of the Mons basin but not on the La Louvière-Centre Basin and Charleroi Basin.

that had the most important impact in the area is the $M_W = 6.0$ 18 September 1692 earthquake that occurred in the Belgian Ardenne (Fig. 1). This large earthquake caused significant damage in the city of Mons where "many houses, churches and other buildings were damaged and half ruined and more than 80 people were either killed or injured" (Alexandre et al., 2008). All these observations suggest that the contribution of the Hainaut coal area seismicity on current seismic hazard maps in Belgium

and northern France (Fig. 1) could be overestimated inside but especially outside the basin and would need to be reevaluated. However, inside the coal area, we have to take into account that the maximal intensity was reported in some localities more than one time.

## 7.2 Intensity attenuation modelling

As our intensity attenuation law provides the only possible substitute for the lack of ground motion data for testing the most

adequate GMPEs for computing seismic hazard, it is important to evaluate if the spatial resolution of the IDPs is sufficiently high for tackling this issue. Indeed, quantifying the rapid intensity decay with epicentral distance needs high-resolution IDP





sampling. Figure 10 clearly shows that intensity decreases fastly by two grades in a range of distances from a few to maximum 10 km from the epicentre. The dataset used to model intensity attenuation contains IDPs that are based on the information from the ROB official survey. From these reports intensity is evaluated by averaging the effects on the overall territory of the different communes for which the authority filled the ROB questionnaire. Before the community fusion in 1977, the size of the communes ranged between 3 and 15 km$^2$, with a mean equivalent circular radius ranging between 1.0 and 2.0 km. After 1977, community size and radius increased and ranges, respectively between 17 and 65 km$^2$ and 2.3 to 4.5 km. The small dimension of the communes explain why the steps considered in the intensity distance binning is 2.5 or 3 km, which are just at the limit of undersampling a range of two intensity values from the epicentre for events with a focal depth of 1 km (Fig. 10). The intensity averaging process in the communes induced by this inquiry also leads to under-estimation of peaks of intensity at local places, an unfortunate effect that is even larger for the bigger communes after the fusion. For some larger earthquakes, we could rely on press reports and letter testimonies to highlight some of these locally increased intensities and to identify where they are located. For Belgian earthquakes between 1977 and 2002, this communal resolution problem complicates intensity modelling. Fortunately, the availability of the ROB online *Did You Feel It?* inquiry since 2002 (Camelbeeck et al., 2003; Lecocq et al., 2009) can resolve this granularity as street addresses of testimonies can be geocoded and intensity data can be aggregated in size-adaptable grid cells (Van Noten et al., 2017). For potential future events, this strategy might allow oversampling the macroseismic field and modelling the intensity variability in each commune, except in localities with extensive damage (cf. the Doughnut Effect in Bossu et al., 2017) where field surveys would then be needed (as done by Sira, 2015).

From the intensity modelling developed in this paper, we now can model the attenuation of ground motion in the coal mining area of Hainaut as follows:

$$I = I_0 - 3.42 * log(\sqrt{\frac{R^2 + Z^2}{Z^2}}) - 0.054 * (\sqrt{R^2 + Z^2} - Z) \tag{6}$$

with $I_0$ determined from the magnitude (see eqs. 3 and 5) or $I_0$ = Imax for earthquakes with only few IDPs, but with a clearly determined epicentral intensity. For these events, Imax scaled to the magnitude (eqs. 4 and 5) can be used for intensity modelling. Applying this attenuation formula (Fig. 7) shows that the intensity prediction works well inside the Hainaut coal area. However, the formula is not meant to predict intensities outside the coal area. Within the border of the Brabant Massif, the e.g. 1967 Carnières event is felt farther than the intensity attenuation model predicts and a different attenuation model should be constructed.

### 7.3 Focal depth determination

Inferring focal depths from macroseismic data provides a robust and alternative way if instrumental data is lacking (Sbarra et al., 2019). Previous authors used intensity data to evaluate the focal depth of some of the largest earthquakes in Hainaut. Charlier (1949) evaluated the focal depth of the 3 April 1949 earthquake to 3.4 km, while Van Gils (1966) provided values of 6.5 km, 4.3 km and 5.0 km respectively for the earthquakes of 15 December 1965, 16 January 1966 at 6h51m and 12h33m. Ahorner, L. (1972) estimated the focal depth of the 15 December 1965, 16 January 1966 and 28 March 1967 respectively to 2.4, 1.9 and 3.0 km. Even if these determinations indicate that these earthquakes occurred at shallow depth, the difference by





a factor of two in the evaluated focal depths between Ahorner, L. (1972) and Van Gils (1966) is difficult to interpret because none of these two authors provide an uncertainty on their determination and explain how they choose the attenuation parameters they used. The approach developed in this study solves these two issues (i) by evaluating attenuation parameters directly from the Hainaut intensity dataset and (ii) by providing a way to evaluate uncertainties linked to the attenuation model and the intensity determination in a systematic way for all the events. Our results show that focal depth estimated by Charlier (1949)

and Ahorner, L. (1972) are inside our error bars.

The ideal test of the robustness of the macroseismic method to evaluate the focal depth of shallow earthquakes would be to compare focal depths determined by this method with the ones estimated by the classic microseismic method based on seismic phase arrival time measurements. In our dataset, the only earthquake for which focal depth was determined from arrival phase measurements in seismic stations is the 8 November 1983 Liège earthquake. In their comprehensive study of the earthquake,

Ahorner, L. and Pelzing, R. (1985) evaluated the focal depth as 6 ± 2 km. Faber and Bonjer (1985) interpreted depth phases recorded by the Gräfenberg network in Germany and concluded that a depth of 4 km would fit better the seismograms. If we use the new Hainaut attenuation model that would be similar in the Liège area, the focal depth of the Liège earthquake is 5.7 ± 1.5 km (see Table 3), which agrees well with instrumental evaluations.

As since 1985, it is possible to evaluate focal depth of earthquakes occurring in the Hainaut coal area by using phase arrival

times of the Belgian seismic network, we compared the depth distribution of the earthquakes that occurred before and after this date to analyse and explain their similarities and (or) differences (Fig. 14).

Since 1985, 29 earthquakes in Hainaut have been located (Fig. 14A) with a depth uncertainty of less than 4 km (Fig. 14B). The largest observed magnitude between 1985 and 2020 is 2.6. Despite a dense seismic network in or near the Hainaut coal area, the focal depth uncertainty still remains significant with a mean value around 2 km, while our estimate of the uncertainties

for earthquakes before 1985 using macroseismic data are lower than 2 km (Fig. 14B). The main reason for this difference is that the distance between earthquake epicentres and the closest seismic station is often greater than 10 km, which is not sufficient to determine focal depths of less than 4-5 km with a high precision (Gomberg et al., 1990).

The two depth distributions coincide for focal depths between 1.75 and 4 km with 24 events on a total of 41 before 1985 and 9 events on a total of 29 after 1985. The two distributions also present two main differences. Before 1985, many events

occurred at very shallow depths of less than 1.75 km (21 events on a total of 41), versus none after 1985. Moreover, most (20 of 29) of the events after 1985 occurred at depths greater than 4.0 km, up to 13 km, while only 3 earthquakes before 1985 occurred at more than 4 km, but still less than 6 km.

All the very shallow events at less than 1 km occurred before 1960, which precedes the end of the mining activities at the end of the seventies. These events contributed only little to the seismic energy release in the Hainaut coal area (Fig. 14C) because

even if most of them were strongly felt or caused slight damage, they were of small magnitude. This is confirmed by the fact that they were not recorded by the seismic station in Uccle (at 35 km for the most northern Hainaut event). Their location inside the coal mining area, their period of occurrence, their very shallow depth and their weak radiated seismic energy could be indicators of a very close link to mining activities.





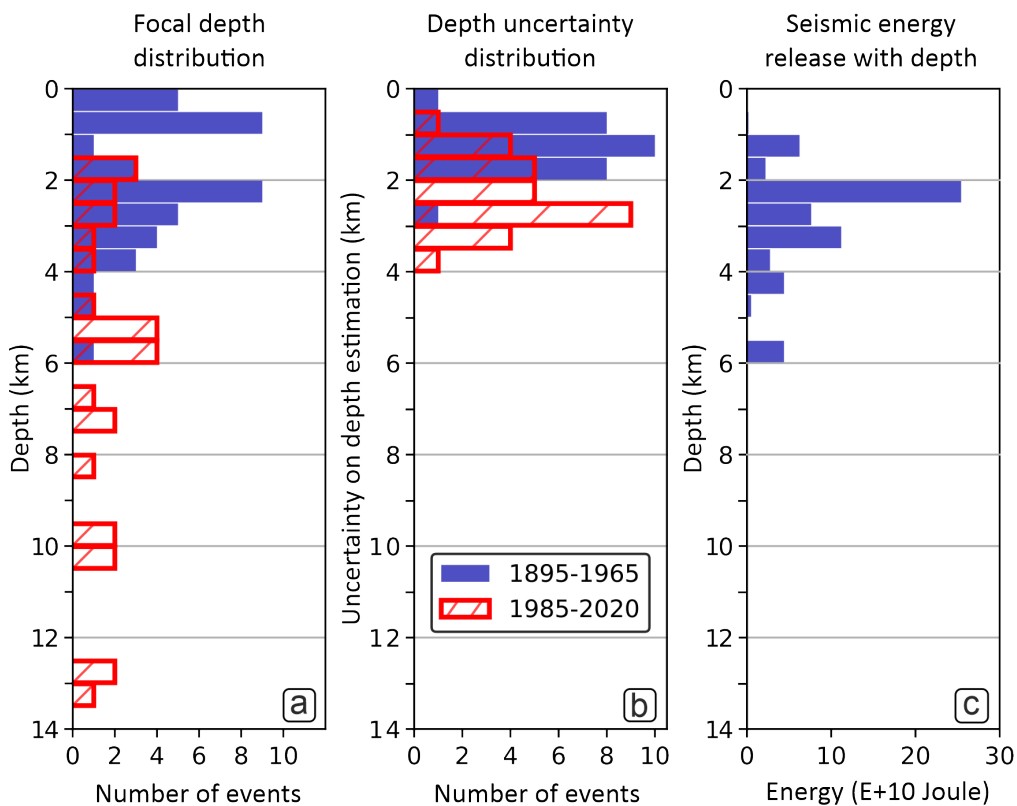

**Figure 14.** A) Focal depth distribution of earthquakes in Hainaut before and after 1985. For earthquakes before 1985, the estimations come from macroseismic data as explained in section 6, while after 1985 depth comes from microseismic location (source: ROB earthquake catalog). B) Distribution of the uncertainties on these focal depth determinations. C) Seismic energy release with depth.

The seismic activity between 2 and 4 km depth, which is below the deepest mining excavations at a little more than 1 km, can

not be directly associated with mining. Nevertheless, the seismic activity strongly diminished after the progressive closure of the mining industry during the seventies, after the high level of activity observed between 1965-1970. This led to the hypothesis that this part of the Hainaut seismicity could be triggered by mining activity. However, the origin of this seismicity should be interpreted at the light of recent studies on earthquake activity in stable continental regions suggesting that it can be explained by transient disturbances of the local crustal stress or changes in fault strength (Camelbeeck et al., 2013; Calais et al., 2016).

Similar questions also arise for the seismic activity deeper than 5 km that has only been observed since 1985. However, the small magnitude of these events could explain that similar earthquakes could have occurred before 1985 but were not detected because they were not felt, nor recorded by any seismic station. These earthquakes could be a background of natural seismicity, but also a seismicity indirectly triggered by the past mining industry. These issues would need to be studied using more quantitative data on stress modifications caused by mining exploitation in the upper crust, time and spatial evolution of

the observed seismicity, earthquake fault-plane solutions, and better interpretation of the surrounding seismotectonic context.



## 8  Conclusions

Our study provides a comprehensive overview of the earthquake activity in the Hainaut coal area and discusses its impact from the end of the 19$^{th}$ century up to 1985, when the implementation of a modern digital seismic network began in Belgium. We updated the ROB earthquake catalogue for magnitude, depth and maximal observed intensity. We also present a digital archive

describing the effects of these earthquakes. We re-evaluated the local intensities of the well-documented earthquakes from these records. They are all included in the Supplement attached to this paper. Our earthquake analysis and impact estimation underline the severity of the damage locally caused by the strongest earthquakes in Hainaut. For earthquakes in the M$_W$ magnitude range between 3.5 and 4.0, maximal observed intensity reaches VI or VII in the EMS-98 macroseismic scale.

Our analysis provides new perspectives for seismic hazard assessment in Hainaut by three aspects. First, it demonstrates

the importance of developing a GMPE for the Hainaut area that is more in line with the observed rapid intensity decay with distance than the current existing European GMPEs. The presented intensity dataset will help to identify the most adequate GMPE. Second, the potential causality between the coal mining extraction that ended in the 1970s and the Hainaut seismicity can now be studied using the new reliable focal depths estimated from the IDP distributions. Finally, the damaging character and the strong intensity attenuation of shallow Hainaut events should be included in the ground motion modelling of potential

induced seismicity related to current and future deep geothermal projects in the area.

**APPENDIX: Intensity evaluation**

**Background to evaluate intensity**

An optimal dataset would be the one describing the way many people in each locality felt an earthquake inside its perceptibility area and furnishing the specific degree of damage for each building hit by the event. This can be obtained when a specific

inquiry is dedicated to collect such a large amount of information. This level of quality is obtained by the ROB online *Did You Feel It?* inquiry since 2002 (Camelbeeck et al., 2003; Lecocq et al., 2009), but up to now, it concerned earthquakes where mean maximal intensity did not reach intensity V in any locality. For intensities equal or larger than V, such an extensive dataset only exists for the destructive 8 November 1983 M$_W$=4.6 Liège earthquake in east Belgium, but this is an exceptional case in NW Europe. This precise damage information came from the owners of 17,000 buildings that sent detailed damage reports of their

property, which was evaluated by the Belgian Federal Calamity Centre in order to reimburse the repair costs. These data were at the base of seismic risk studies on the Liège area (Jongmans and Plumier, 2000; Garcia Moreno and Camelbeeck, 2013; Camelbeeck et al., 2014).

The ROB survey and some of the scientific studies described in section 3 are not so detailed, but they furnish information to evaluate intensity at the scale of each locality and have the advantage to sample the complete macroseismic field of the studied

earthquakes. Information in the press does not sample the whole area of perceptibility and is often concentrated on the most visible effects of the earthquakes. We determine intensity in the following way: when the answers to the questions in the ROB questionnaire and (or) information from other sources fulfil and exceed the EMS-98 description of the earthquake effects at a





given intensity degree $I$, but are not compatible to the description corresponding to a higher intensity value $I+1$, the intensity is fixed to the single integer value I. When the observations do not allow discriminating between two intensity values, a range of corresponding intensity values is given. Information coming from some localities for earthquakes that were not the object of an official survey is sometimes insufficient to assess intensity although the seismic event was reported as felt. We indicated these places with an "F" on the macroseismic maps. When the answers to the ROB official survey in one locality were all negative (see inquiry books in the Supplement), we considered the earthquake as not felt there, but we do not report this information on the macroseismic maps as the consulted sources are insufficient to establish the limit of perceptibility.

**Building vulnerability**

For intensity greater or equal to V, a significant part of our evaluations comes from damage observations. To assess intensity, it is necessary to know the building stock and vulnerability class distribution in the studied area from the beginning of the $20^{th}$ century to around 1970. At the exception of Barszez (2005), who studied the seismic vulnerability of historical houses in the centre of the Mons, there is no study analyzing the seismic resistance of buildings in the Hainaut coal area. Fortunately, the building stock is relatively similar to the one in the Liège region that was well studied after the 1983 Liège earthquake (Garcia Moreno and Camelbeeck, 2013; Phillips, 1985; Plumier, 1985, 2007). The main reason for this resemblance is that the two regions experienced a similar rapid population expansion due to strong industrial development that accompanied the extensive exploitation of coal and development of an important steel industry. Unreinforced masonry houses formed an important part of the building stock, which was common in this part of Europe during the $20^{th}$ century. This type of building is associated with vulnerability class B in the EMS-98, but it can range between class A for the most vulnerable and class C for the least vulnerable buildings according to the quality of their foundation, construction and maintenance.

During the 1983 Liège earthquake, part of these masonry buildings showed deficiencies which were at the origin of serious structural damage. The most affected structures were unreinforced low-rise masonry dwellings for which the links of the floors and the load-bearing walls were weak or even missing. Many of those buildings shared walls with the neighboring houses (Phillips, 1985; Plumier, 1985, 2007). The importance of the damage on these buildings compared to the better behaviour of well-constructed brick buildings clearly suggest that they belong to class A in the EMS-98 classification. In the Hainaut coal area, the same type of buildings are represented in many corons where families of workers in the mining and siderurgy industries are living. However, many buildings also suffered from damage directly associated with mining activities including the underground progression of coal exploitation and the progressive settling that follows (see discussion). Increased humidity due to surfacing groundwater and pre-existing structural weaknesses associated with mining activities increased the vulnerability of buildings. These aggravating circumstances suggest that a significant part, which is unfortunately undetermined, of the building stock are to be classified in the class A vulnerability defined in the EMS-98 macroseismic scale.

**Intensity from damage**

In the ROB questionnaire, questions concerning damage to buildings allow fixing intensity equal to or greater than V (see inquiry books). The observation of small fragments of plasters that fell from the ceilings and of broken or cracked windows



appear at intensity V. EMS-98 considers brick chimneys behaviour as representative of the damage grade on masonry buildings because it is the most visible manifestation of the seismic action during moderate earthquakes. Indeed, fireplaces are slender objects, not very resistant to bending, especially since the corrosion of the mortar transforms them into a pile of bricks stacked without much connection (Plumier, 1985). Their partial collapse is an indicator of damage grade 2 (moderate), while fracture

at the roof junction corresponds to grade 3 (sensitive to severe damage). The last question in the form asks the local authorities about the number of damaged and overturned chimneys, which theoretically allows the seismologist to evaluate the percentage of grades 2 and 3 damage in the locality. Considering that the most important damage occurred on the most vulnerable part of the masonry buildings, the quantity of fallen/damaged chimneys provides a way to either confirm intensity V (very few damaged chimneys) or help discriminating between intensity VI and VII if, respectively, few or many chimneys were overturned.

The EMS-98 scale defines the limit between the quantities "few" and "many" as being between 10% and 20% of the number of considered buildings in a specific vulnerability class. Then, the percentage of building vulnerability class A in a locality is an important factor in the intensity evaluation process. Unfortunately, this information is lacking and we are forced to make simplistic assumptions about it. Here, we considered that half of the buildings are in class A and that only these most vulnerable structures suffered the highest observed damage grade. This means that the observation of 5% or more of overturned chimneys

in a locality would correspond to 10% or more of grade 3 damage on vulnerability class A buildings, which corresponds to intensity VII. Of course, grade 2 damage should also be observed in many buildings of class B and class A. We considered that simultaneous observation of 5% grade 3 and 5% grade 2 damage would be associated with intensity VII. When both of these percentages of damage are smaller than 5%, we assign intensity as VI if they are greater than 1% and V if they are smaller. The official survey also asks for the observation of large and extensive cracks in walls. A positive answer to this question indicates

damage grade 3, but as the question does not ask for any quantification, it is not possible to fix the intensity to VI or VII based on this information.

At intensity VII, reports should mention serious failure of walls and partial structural failure of roofs and floors, corresponding to grade 4 damage, in few buildings of class A. Unfortunately, the ROB questionnaire does not allow to identify the importance of cracks in walls and building structural damage. Assessing this kind of damage would require specific building

inspections by a specialised engineer. Nevertheless, press articles provided local observations that we interpreted as grade 4 damage and can be used to confirm the estimated intensity of VII in some localities.

*Code availability.* Codes are available from the authors upon request.

*Data availability.* Earthquakes that occurred in Hainaut are included in the entire earthquake catalogue maintained by the Royal Observatory of Belgium (ROB) and which can be consulted online at http://seismologie.be/en/seismology/seismicity-in-belgium/online-database



*Supplement.* The supplement related to this article is available online at https://doi.org/10.5194/nhess-0-1-2021-supplementnd includes:

- Table S1: The Hainaut earthquake catalogue provided as csv file (123 events)

- The Hainaut Intensity Atlas which presents

  - the Hainaut seismicity catalogue (123 events);

  - 31 intensity maps of 28 Hainaut events and 3 additional earthquakes that had a large impact on the Hainaut coal area;

  - 12 intensity-distance modelling graphs;

  - Sources and references for the entire catalogue

- 28 csv files containing intensity data of the earthquakes mapped in the Atlas

- Forms of each municipality that replied to the official surveys of the Royal Observatory of Belgium. For 17 earthquakes, an intensity
  inquiry book is made available in pdf.

*Author contributions.* TC conceptualised the research, constructed the earthquake catalogue, evaluated local intensities and executed the
intensity attenuation, focal depth and magnitude modelling. TC and MH searched for historical texts in press reports. KVN produced the
(intensity) maps and the Atlas. TL computed and summarised the official ROB intensity surveys. TL and KVN calculated the cumulative
Hainaut impact. TC, KVN and TL wrote the paper. All authors approved the final paper.

*Competing interests.* The authors declare that they have no conflict of interest.

*Acknowledgements.* We thank Pierre Alexandre, David Kusman and Rita Leroy for their search of documents concerning the earthquake
activity in the Hainaut coal area. All authors were funded by the Royal Observatory of Belgium. Data analysis and modelling was performed
using Python, using Scipy (Virtanen et al., 2020), Numpy (Oliphant, 2006), Geocoder (Carriere, 2021), and Pandas (McKinney, 2017)
modules. Graphs were created using MatPlotLib (Hunter, 2007). Maps and attenuation modelling were constructed using QGIS (QGIS
Development Team, 2021). Population density statistics for Belgium were kindly provided by Anne Joris of Statbel (Algemene Directie
Statistiek – Statistics Belgium: https://statbel.fgov.be/en). The geological background in Figs. 1, 2, 3, 4, 5, 6, 7 and in all macroseismic
maps in the supplement origin from http://www.onegeology.org/. Reproduced with the permission of the OneGeology. All rights Reserved.
Newspaper images shown in the supplement are free of copyright according to https://www.kbr.be/en/digitisation/digital-collections-what-
about-copyrights/.

*Financial support.* This research was supported by funding of the Royal Observatory of Belgium.





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
