# Peer review of "The damaging character of shallow $20^{th}$ century earthquakes in the Hainaut coal area (Belgium)"

_Solid Earth, 2021_

## Author Comment (AC2)

**SUPPLEMENT - REPLY TO THE REVIEWER 1: Figures**

[Figure]

[Figure]

[Figure]

*Figure that will be added to the paper:*

*Comparison of binned (3 km) intensity - distance observations for*
*a) MW3.5 earthquakes in Hainaut, in 2012 in Huizinge (Groningen gasfield, NL, Dost and Kraaijpoel, 2013), and in Oklahoma (US, Atkinson, 2020),*
*b) Mw4.0 earthquakes in Hainaut and in Oklahoma (US),*
*and c) Mw5.0 earthquakes in Le Teil ((FR, Sira et al., 2019), Völkerhausen (DE, Leydeckeret al., 1998) and Oklahoma (US).*

*IDPs (small grey dots), mean intensity (coloured dots or squares) and standard deviation (bars) of the different intensity datasets are shown in comparison with the fast decay of the Hainaut intensity attenuation relationship (green line). Z =depth in kmIt shows a comparison between Hainaut attenuation (green line = attenuation law, blue dots = distance bins) and Oklahoma (US) and Huizinge (NL) Mw3.5 and Mw4 events.*

From this comparison we can conclude that for a shallow Mw3.5 event negligible to slight damage (starting from I=V) for M3.5 at the epicenter until a mean distance of 3 km or 4 km for the 84 percentile. In Oklahoma, this observation is relatively similar.

[Figure]

For shallow Mw4.0 events, substantial damage (I = 7) will occur at the epicenter until a median distance of 2 km and 4 km for the 84 percentile, moderate damage for magnitude M4.0 (starting from I = VI) until a median distance of 3 km (7 km 84 percentile) and negligible to slight damage until a median distance of 5 km (8 km 84 percentile).

[Figure]

**Section 7.2:**

[Figure]

**Figure:** *Azimuthal analysis of the ML 4.5 1967 Carnières earthquake. Intensity values are separated in 20° azimuthal bins. This azimuthal analysis was performed for all 12 earthquakes that were used for modelling.*

---

## Author Response (AR1)

Dear Editor,

Thank you very much for the taken to edit our paper entitled:

**The damaging character of shallow 20th century earthquakes in the Hainaut coal area (Belgium)**

In the interactive stage of the review we gave a detailed reply to the three reviewers. In the letter below, we summarise these detailed replies again. Reviewer 2 questioned the interest of this paper towards an international audience but we replied to that comment by presenting all the scientific issues that are timely and of interest to a wide public. The reviews encouraged us to make substantial changes to the text.

- First two reviewers requested to move the section on the detailed description of the earthquakes to the Supplement. We prefered moving the description in an Appendix B so that the examples remain in the paper as reference material. We left one figure in the paper (Fig. 3 now) that shows what the earthquake maps look like;
- As there was confusion on when the seismicity started, we added a paragraph in Section 2 explaining why we express that no significant (felt or damaging) seismicity occurred in the beginning of the coal mining;
- As requested by R1, we added a paragraph in Section 3.3 on the macroseismic survey books that provided in the Supplement;
- Despite some small textual changes, we changed nothing on the methodology of the attenuation modelling in Section 4;
- Following R1's request, we added a *Section 5.1 How Fast is the Hainaut attenuation?* to explain the difference between our modelling results and other regions worldwide that are subjected to induced or triggered seismicity. We are certain that this comparison will be of international interest;
- Following R1's request, we added a paragraph in Section 5.2 adding an explanation on the population density, the spatial distribution of intensity points in the Hainaut coal area and how the regional geology affects this intensity distribution.
- All reviewers requested to highlight our findings on the need to e-evaluate the hazard modelling. We followed this advice and highlighted this much more in the abstract, introduction and conclusions.

We hope that you can agree with all these changes and that you will consider this re-submission as a valuable contribution to the journal *Solid Earth.*

Kind                                                                                                                          regards
Thierry Camelbeeck for the authors

*Section 4*

*I feel the discussion of individual cases in the middle of the paper, i.e., all of section 4 might be better suited for the Supplementary Material as opposed to within the paper. I understand that the authors wish to draw attention to each of these earthquakes. However, it does disrupt the flow of the paper – there are 11 pages in total. A suggested workaround might be to move a substantial number of these events into the Supplementary Material, retaining only a small handful within the text of the paper that stood out for some reason.*

**We indeed want to draw the attention to these widely felt or damaging earthquakes as these were never reviewed in detail in the past. Yet, as the reviewer indicates, it disrupted the flow of the paper. We followed the reviewer's advice and moved both description and the figures to Appendix B. We prefer the Appendix over Supplementary material so that the examples remain still in the paper and can be used as reference material while reading the paper.**

**Section 4 now became Appendix B and we kept a short descriptive paragraph in *section 3.4 Intensity database* to explain the appendix and the figures. We also left one map in the paper (Figure 3 now) to show how the maps look like and how intensity data points fit the modelled attenuation for one event.**

*Supplementary Material*

*I am happy to see tabulated lists of intensity assessments for each event. This involves a lot of work, not to mention the hours/days/months in archives battling dust and maybe spiders…… These lists (and future ones) will benefit from the inclusion of summarized effects for each location with a reference (or references). This will seem like a big ask to incorporate at this stage in the project and does not impact the outcome of this article! It is a recommendation only; one I hope that future work of this nature will incorporate.*

**Thank you for the appreciation of our work on the supplementary data. You are right that adding the effects of each location with a reference would even more clarify the process from observation to intensity evaluation. However, that's indeed a big task at this moment but we will take this comment with us to the future when we evaluate the macroseismic database of Belgium (work in progress). Nevertheless, thanks to the reviewer's comment, we highlighted the importance of the intensity books that are provided in the Supplement.**

*Section 7.1*

*It might be useful to consider the decay of intensity from other induced events outside the Hainault area. Perhaps put this in context with any similar work that might exist from Groningen gathered by the Netherlands version of the DYFI? Koen should have access to these if this is an avenue you decide to pursue. Alternatively, I think there is some work done for induced events in the central US and Canada where there is a distinct intensity signal between natural events and induced earthquakes. You might wish to consider papers by Gail Atkinson and Susan Hough who have looked at these more closely, and then put your work from Hainault in context from observation. How does it compare? Or does it not compare?*

[Figure]

[Figure]

[Figure]

To discuss the difference between the Hainaut attenuation and other worldwide shallow events, we followed a similar approach as Atkinson (2020), who discussed the damaging character of Oklahoma triggered events using the USGS DYFI? database. In our comparison, we gathered the intensity data of all M3.5±0.1 (n=3) and M4.0±0.1 (n=4) Hainaut events and computed the mean ± std intensity decay using distance bins of ~3 km through our intensity data. Then, we compare this decay with the median distance classes from Atkinson 2020 and our modelled Hainaut attenuation. For M3.5 events, we also added the intensity data of the Huizinge2012 (Z=3 km) event in the Groningen gas field (NL) that caused local damage.

The comparison of the Hainaut attenuation with Huizinge and Oklahoma events shows that the computed Hainaut attenuation (green line) is very fast. This observation indicates that the shallow depth mainly explains the damaging impact in the Hainaut region and not the attenuation. Oklahoma events are usually deeper than events in Hainaut (Z ~5 km, Atkinson 2020), showing e.g. that a M4.0 in Oklahoma will have more impact in terms of damage than in Hainaut.

For M5 events, we compared the Hainaut attenuation with intensity data of the Völkerhausen 1989 (DE) potassium and salt mining triggered event (0.8 km) and the 2019 Le Teil (FR; 1 km) event (Schlupp et al. in review). Here, the observation is similar and the Hainaut attenuation is again much faster than in these areas.

*Figure added to the paper showing a comparison between Hainaut attenuation (green line = attenuation law, blue dots = distance bins) and Oklahoma (US) and Huizinge (NL) Mw3.5 and Mw4 events.*

From this comparison we can conclude that for a shallow Mw3.5 event negligible to slight damage (starting from I=V) for M3.5 at the epicenter until a mean distance of 3 km or 4 km for the 84 percentile. In Oklahoma, this observation is relatively similar.

[Figure]

For shallow Mw4.0 events, substantial damage (I = 7) will occur at the epicenter until a median distance of 2 km and 4 km for the 84 percentile, moderate damage for magnitude M4.0 (starting from I = VI) until a median distance of 3 km (7 km 84 percentile) and negligible to slight damage until a median distance of 5 km (8 km 84 percentile).

[Figure]

The Hainaut attenuation is fast because of (i) coal deposits are located in the frontal zone of the Variscan belt where the subsurface is strongly fractured during Variscan compressional tectonics and (ii) the combined effect of a high fracturation degree and the low Q-factor associated to the slow propagation velocity of coal. This results in a faster attenuation compared to the neighbouring regions such as the Brabant Massif or Ardennes, where bedrock with high propagation velocity is present near the surface.

*The authors should begin this section by noting whether or not the spatial distribution of intensity observations is adequate in number and is spatially unbiased. I draw attention to Meltzner & Wald (1999) and Hough & Martin (2018) who comment/show how the number of points/observations and their azimuthal distribution can produce biased results. Admittedly, you do not use the modelling approaches that were later used by both those studies, but I feel this is an important point to make. I believe, Van Noten et al. (2017) shows how intensities can be biased by subsurface geology and Hough & Martin (2021) show how intensities, particularly, historical intensities can be biased by social factors.*

*Can you add a few lines to reassure the reader that your intensity data, and by that measure, the attenuation relations are not biased by geology, social factors, and spatial distribution of observations? How reliable are the early reports of shaking effects in the mining areas? Is there the possibility that local mining companies might have wished to keep reports of shaking and or damage hidden from public view?*

You are right in making this comment but the spatial and azimuthal distribution is constant within the Hainaut area. Within the Hainaut coal mining area, the population density is quite constant and dense in the cities (see inlet in the macroseismic maps, density ranging often between 500 and 2500 inh/km²). The official forms that returned to the Royal Observatory of Belgium are statistically valid as they contain a summary of e.g. police, firemen and communal reports. So the data gathered by the communities is adequate in number. Outside the Hainaut area (within the border of the Brabant Massif and Ardennes), population density is lower and areas are more rural. This geographical difference explains why a lower amount of responses was returned to the ROB from these areas.

To examine the possibility of an azimuthal underrepresentation of the data, we composed an azimuthal intensity analysis (see figure below) for each of the 12 events that are used for attenuation modelling. Within each azimuthal bin (20° bins), we have intensity data, hence the azimuthal coverage was sufficient to construct our attenuation law. For smaller events, this was not always the case, but this data was not used in the attenuation modelling. The large difference between the North-South maximal perceptibility in the example below is dominantly linked to the local geology, rather than to changes in azimuthal distribution. The 1967 Carnières event was wider felt in the Brabant Massif to the north than in the Ardennes in the south. So yes, intensity distribution is biased by subsurface geology (as indicated by Van Noten et al. 2017), but within the small Hainaut coal mining region we have evidence that using a homogeneous intensity distribution for modelling is justified.

Within our intensity evaluation we indeed also used reports from the mining companies. However, these are just a minority of observations. The *"Felt in the mines"* observations (see double red stars in the inlet in various macroseismic maps) is sourced from newspaper reports and often do not come from mining reports directly. The surface damage during the largest events was large and can be deduced from the community reports and hence the mining reports were not needed for estimating the impact. So there is only a small risk that mining companies would have hidden shaking observations for themselves.

[Figure]

*Figure: Azimuthal analysis of the ML 4.5 1967 Carnières earthquake. Intensity values are separated in 20° azimuthal bins. This azimuthal analysis was performed for all 12 earthquakes that were used for modelling.*

*Section 7.3*

*Here the authors draw the conclusion that the shallowest seismicity was triggered by mining in the region and ended with the cessation of mining. However, nowhere in the article it is discussed when this seismicity began AND what was the local seismicity prior to the start of mining for the period in question. Can these two questions be answered without returning to the archives? If you do this, I feel it would add further weight to your assumption (Lines 592 – 593) that the very shallow seismicity was indeed triggered.*

*Thank you for this valuable comment. However, neither in our archives, nor in scientific and press reports there is a mention of seismicity that could have occurred prior to the start of the seismicity in our catalogue. This does not mean that low-magnitude seismicity never occurred (see also our reply to reviewer 2), but it seems very unlikely that damaging or widely-felt events would have been overlooked in the press reports and in the existing scientific work. In Belgium and surrounding regions, historical events are well catalogued and described in newspapers (e.g. Alexandre et al. 2007, Knuts et al. 2016, Camelbeeck et al. 2021), hence, this makes us confident to state that local widely-felt or damaging seismicity started after 1887.*
* * *
*Summary*

*R2: This paper presents an exhaustive revaluation of 28 small earthquakes in Belgium between 1887 and 1983 in the Hainaut province of SW Belgium. Macroseismic data are presented in detail and used to refine parameters.*

*Comments*

*R2: Technically, this paper is of high quality, with only one error, which is to refer to "attenuation laws" for things that are not laws.*

**We sincerely thank the reviewer for the general comment about the quality of our research. We agree with the comment and changed the term *intensity attenuation law* into *intensity attenuation relationship* everywhere in the paper.**

*R2: The problem with this paper is that it calls to mind the apocryphal most boring headline ever – "Small earthquake in Belgium: No-one hurt". This paper is 40 pages long in the preprint version, and it is questionable what interest it has for an international audience such as the readership of Solid Earth. There is no doubting the quality of the research, but one would normally expect it to appear as a published report of the sponsoring institute rather than a paper in an international journal.*

**Following both reviewer's comments (see also our reply to reviewer 1), we reduced the length of the paper by transferring the description of the seismicity (previous section 4) to Appendix B in the paper.**

**The reviewer furthermore questions the interest of our submitted manuscript for an international audience such as the readership of Solid Earth because it only concerns small damaging earthquakes in Belgium. The submitted version of our paper was indeed partly concentrated on the regional context of seismicity in Belgium, but it also focused on aspects that are of international interest:**

**1. We provide a strict methodological framework concerning the way an historical/early-instrument earthquake catalogue should be composed and improved based on re-evaluation of historical sources. In the electronic supplement, we provided all references to historical sources and the intensity database which allows the reader to re-evaluate the whole work. Coupling the data with the analyses encourages researchers to build their own opinion on the problematic and motivates them to apply a same methodological approach for other areas where similar moderate past seismicity occurred. Moreover, the Solid Earth journal promotes sharing new data in their open repository to make the work reproducible.**

**2. Even though the shallow earthquakes may have looked small, the findings of our paper (attenuation, depth, etc) have important implications for decision-makers related to subsurface management. The sensitivity of highly populated areas to small, shallow, induced and triggered, potentially damaging seismicity strongly increased during the last years. Therefore it is important to evaluate if past seismicity considered as natural could be related to human activities.**

**3. To increase the interest in the work, we added a comparison of the attenuation intensity and impact between Hainaut and other regions worldwide subjected to induced or triggered seismicity, such as in Groningen, Oklahoma, Völkerhausen, Le Teil, etc. (please see our detailed reply to reviewer 1).**

*R2: The significance is given in the inset to Figure 1, showing the Hainaut area being depicted as of high seismic hazard in the recent SHARE European hazard map. From the authors' work this is clearly incorrect.*

*Most of the earthquakes in this region are too small to be hazard-relevant, and probably all of them are mining-related, in which case they should be deleted from the earthquake catalogue prior to PSHA.*

**Absolutely, this was part of our motivations for conducting this work. Before this study, the origin of the earthquake activity was considered as natural. The activity was never labelled "induced or triggered". In a country where damaging events are rare, this seismicity is significant, and consequently the whole catalogue was integrated in the different PSHA studies. Of course, even though events were small, they were hazardous because the events were recurrent and caused damage (as shown in the discussion). With our work, re-evaluation of PSHA is encouraged (and is actually ongoing).**

**Based on the reviewer's comment, we highlighted this aspect more in the abstract, introduction and conclusions of the paper.**

*R2: It is significant that despite the frequency for events in the period under consideration, there is a complete lack of earthquakes prior to 1887, which is a clear sign that there is no natural tectonic seismicity.*

**Recent work on Stable Continental Regions show that seismicity is sporadic and moves in time and space. The absence of seismicity in SCR for a given period of time is not necessarily a sign of absence of natural activity.**

*R2: To make this paper of interest, the focus should be shifted away from the detailed data on these earthquakes (all of which can be moved to an ROB report) and to the more general topics of mining seismicity and hazard. There is at present no examination of what was done for Hainaut in the SHARE project – for instance, how were these events depicted in the SHARE earthquake catalogue, compared to the authors' final versions? This seems like an obvious topic to cover. Also, it would be useful to have a history of coal mining in the region to compare to the progress of the Hainaut earthquakes. What is the likely effect on the PGA hazard to be expected from deleting all these non-tectonic events? These are the sorts of questions that ought to be covered in a paper for Solid Earth.*

**In the EMEC catalogue used in SHARE (Grünthal, G. & Wahlström, R., 2012), Mw was evaluated for most events from the Belgian catalogue from Imax. These magnitudes are sometimes not correct for two reasons: first, because the depth and local attenuation were not considered in the magnitude evaluation; second, the re-evaluation of intensity in all available macroseismic sources shows over-estimation of some Imax values in the original ROB catalogue used in EMEC. For example the 1953-09-15 event in Quaregnon (id 557), Imax was originally set to I = VII, while our re-evaluation reduced it to I = V. The comparison of how seismic parameters in our catalogue (CAM2021, in red) changed with respect to the EMEC catalogue is shown in the table below:**

| E-ID | Date | Time (catalogue) | Time | Depth CAM2021 | Depth EMEC | intensity CAM2021 | intensity EMEC | M_orig | type | Mw CAM2021 | Mw EMEC |
|---|---|---|---|---|---|---|---|---|---|---|---|
| 431 | 1887-09-20 | 06:40:--.-- | 6:40 | [0.4] | | 4_5 | 6 | | | 1.7 | 4.3 |
| 434 | 1887-10-29 | 21:--:--:-- | 21: | [0.8] | | 4_5 | 5 | | | 2.3 | 3.7 |
| 438 | 1904-04-23 | 16:30:--.-- | 16:30 | [0.8] | | 4 | 5 | | | 2.1 | 3.7 |
| 465 | 1911-06-01 | 22:51:58.-- | 22:51 | 4.3 | | 6 | 7 | 4.2 | L | [3.9] | 3.9 |
| 466 | 1911-06-03 | 14:35:54.-- | 14:35 | [1.4] | | 7 | 6 | 4.1 | L | [4.0] | 3.8 |
| 505 | 1936-11-05 | 00:41:44.-- | 0:41 | [2.2] | | 4_5 | 5 | | | 3.3 | 3.7 |
| 533 | 1949-04-03 | 12:27:38.-- | 12:27 | | | 7 | 6 | 3.9 | L | [3.7] | 3.6 |
| 534 | 1949-04-03 | 12:33:40.-- | 12:33 | 2.2 | | 7 | 7 | 4.6 | L | [4.1] | 4.3 |
| 539 | 1949-04-14 | 05:12:21.-- | 5:12 | [2.4] | | 6 | 2 | 3.8 | L | [3.6] | 3.5 |
| 547 | 1952-10-21 | 21:15:--.-- | 21:15 | [2.9] | | 4 | 6 | | | 3.1 | 4.3 |
| 549 | 1952-10-27 | 06:11:--.-- | 6:11 | 3.5 | | 5 | 6 | | | 3.5 | 4.3 |
| 557 | 1953-09-15 | 23:55:--.-- | 23:55 | | | 5 | 7 | | | 3.1 | 5 |
| 562 | 1954-07-10 | 17:18:21.-- | 17:18 | 3.3 | | 5 | 6 | | | 3.5 | 4.3 |
| 573 | 1957-01-08 | 16:12:--.-- | 16:12 | [0.4] | | 6 | 6 | | | 2.2 | 4.3 |
| 577 | 1958-05-30 | 14:45:--.-- | 14:45 | [0.6] | | 6 | 6 | | | 2.6 | 4.3 |
| 582 | 1965-12-15 | 00:07:15 | 12:07 | 2.7 | | 7 | 7 | 4.4 | L | 4 | 4.1 |
| 588 | 1966-01-16 | 00:51:35 | 6:51 | 3.3 | 5 | 5 | | 3.8 | L | 3.5 | 3.5 |
| 589 | 1966-01-16 | 00:32:50 | 12:32 | 2.1 | | 7 | 7 | 4.4 | L | 4 | 4.1 |
| 595 | 1966-03-20 | 00:08:15 | 0:08 | | 5 | | | 3.8 | L | 3.5 | 3.5 |
| 597 | 1967-03-28 | 00:49:25 | 15:49 | 3 | | 7 | 7 | 4.5 | L | 4.1 | 4.2 |
| 606 | 1968-08-13 | 00:57:14 | 16:57 | 2.3 | | 6 | 7 | 4.1 | L | 3.9 | 3.8 |
| 612 | 1970-11-03 | 00:46:00 | 8:45 | 2.3 | | 5 | 7 | 3.9 | L | 3.6 | 3.6 |
| 627 | 1976-10-24 | 00:33:28 | 20:33 | 5.5 | | 6 | 6 | 4.2 | L | [3.9] | 3.9 |

**Especially for the 19th and early 20th century events, intensity, origin time and magnitude values changed considerably. Our paper explains in a detailed way how these values were changed.**

**Our results show that not only the origin of the activity cannot be considered independent from the mining history, but that also the strong attenuation in the Hainaut Coal Area is an important element to take into account in future PSHA computations for SW Belgium and surrounding regions, i.e. Northern France.**

**The work to correct and extend earthquake catalogues, following the rules of historical criticism, is a huge task. It asks the courage to accept that previous works and catalogues may contain errors and may need re-evaluation. One of our objectives was to demonstrate this statement and to provide the particular ground motion attenuation and seismicity background, necessary to improve hazard mapping in the near future.**
* * *
**REVIEW 3**

*Summary:*

*This review comes after two accurate anonymous reviews which already addressed most of the questions I intended to raise. I will briefly recall my main points.*

*The paper is important and timely, but it is indeed quite long. It looks more like a catalogue, or a technical report, rather than like a scientific paper that extracts something relevant from the data presented. I praise the reviewers' recommendation to move the description of individual events into the Supplementary Materials section, so that the reader is free to focus on the most relevant conclusions of the work.*

**Thank you for this comment. May we refer to our replies of Reviewers 1 and 2 in which we explain how we shortened the paper. Whole section 4 (detailed description of the stronger earthquakes) has been moved to an appendix.**

*But I don't want the authors to get me wrong. I think their work is valuable for two independent reasons: for their effort to reorganize the historical seismicity data in a systematic and objective fashion, and for their attempt to use the new data to improve SHA estimates for Belgium. As a matter of fact, I would urge the authors to consider using the data presented in this work to turn the simple parametric catalogue they run on the ROB website into a fully analytical catalogue such as Italy's Catalogue of Strong Earthquakes (http://storing.ingv.it/cfti/cfti5/#). This would be the best way to make sure that their efforts at investigating old earthquakes live over time.*

**Thank you for the comment and the link to Italy's Catalogue of Strong Earthquakes. This is an amazing and valuable website! It will require some efforts from our side to mimic this website and prepare all our data in QuakeML and its extended version including all macroseismic data, but we will definitely take this comment with us in the future. In the meantime, many of the historical events in Belgium and surrounding areas and their macroseismic dataset are already made available on the ROB seismicity website and in AHEAD for pre-1900 earthquakes (if a scientific work on the event was published). We hope this will also be the case for the Hainaut part of the ROB catalogue.**

*The paper is made even longer by the particular wording adopted by the authors. The English is generally good and correct and I have no complaints (although I also noticed a few cases where the authors use single words/expressions that do not really mean what they have in mind: perhaps they are "false friends"?). I am not a native English speaker myself, but I am convinced that adopting a lighter prose and getting rid of some unnecessary explanations might reduce the total length of the text by 10-20%, in addition to reorganizing the text as detailed above.*

**Yes, you are right. Thanks also to the comments of the previous two reviewers, we re-edited the paper entirely, rephrased many sentences and removed some paragraphs with duplications.**

*All in all, the paper is an important contribution both to the homogenization and reappraisal of historical seismicity data for Belgium and to the quantification of regional seismic hazard, and it certainly deserves to be published on Solid Earth. Nevertheless, I do agree with one of the reviewers who stated that the paper should discuss in better detail the SHA implications of the new findings. I will be more specific in the specific comments listed below.*

**Thank you again for these comments. As explained in our reply to reviewer 2, in the new version of the paper we now highlight the impact towards SHA much more in the abstract, introduction, discussion and conclusions.**

*Specific comments*

*Line 24 - " Hence, because of their shallow sources, moderate SCR earthquakes with magnitudes in the range of 4.0 to 6.0 are often more damaging in SCR than at plate boundaries". I don't share this view. In Italian volcanic areas such as Mt. Etna and the Ischia Island, which certainly do not lie within a SCR, M 4.0 earthquake are capable of causing substantial damage and casualties.*

**We based this assumption on a statistical analysis that the percentage of shallow earthquakes is higher in SCR than at plate boundaries. Of course, this does not mean that shallow M=4.0 earthquakes cannot occur at plate boundaries. We deleted the first sentence you refer to above, and modified the text in the abstract and in the introduction as follows to to take this remark into account:**

**"Moderate shallow earthquakes with magnitudes in the range of 4.0 to 6.0 have a real potential of destruction when they occur in populated areas. This is particularly the case in regions where the building stock is old and vulnerable, and contains few earthquake-resistant buildings. In seismically active regions, even if 4.0 MW earthquakes can be locally damaging (Nappi et al., 2021), current hazard is associated with the upper part of this magnitude range…"**

*Line 163 - "However, as the earthquake occurred at midnight, there was no notice of the event outside a radius of 3 to 4 km from the barycentre of all macroseismic data points". I don' t understand this. What difference does it make if the earthquake occurred at night or during the day? It was felt over a small area because it was shallow, regardless of the time of the day.*

**The reason is that at a larger distance many people were sleeping and were not awakened by the earthquake. This leads to a lack of observations in these areas where you normally would have observed effects associated to intensities II to IV available during the daytime. Hence, it was not possible to evaluate the real extension of the macroseismic field. We modified the text to take this remark into account.**

*Line 168 - "Two months later, the earth shook again north of Charleroi, but more strongly with a MW =3.9 event on 1 June at 22h51m... The epicentral area of the 1 June 1911 earthquake includes the localities of Gosselies, Lambusart and Ransart where the tremors were violent enough to awaken most of the inhabitants, knocking down many chimneys and causing cracks...". The authors should compare this event and its effect with those of the 27 August 2017, Ischia earthquake, a Mw 3.9/ Md 4.0 event located around 1 km depth.*

**Thank you for your suggestion. In the new version of the paper, we compare the Hainaut attenuation model and earthquake effects with macroseismic data of other shallow earthquakes to demonstrate how fast the Hainaut model is (see our detailed reply to reviewer 1 and figures in supplement). We deliberately chose to compare it with Mw 4.0 events from Oklahoma, as these events are recurrent and many data points are provided. These events occur in a similar intraplate tectonic context as the Hainaut seismicity. We believe that adding a comparison with a volcanic context might confuse the reader so we decided not to add the intensity data points of the Ischia earthquake.**

*Line 172 - "...the most affected locality was Ransart where more or less 50 chimneys collapsed and a parked train was thrown off the tracks... We assessed intensity to VI in Ransart ...". Are you extra sure of this statement? Seems to me that overturning a train would require accelerations that are incompatible with the size of this quake, even if it occurred at very shallow depth. And an intensity VI sounds very low for such a damage scenario.*

**The train that was derailed from the tracks was a smaller mine train used by the mining industry. It is difficult to evaluate the stability of such a vehicle on a small mine rail inside an industrial field. The observation suggests high accelerations locally that would likely correspond to an intensity higher than VI, but there is no description of building damage for a mean intensity greater than VI in Ransart. We also clarified in the discussion that the spatial resolution of our intensity data is too low to identify very local intensity effects [see section 6.3].**

**We slightly changed the wording in appendix B to:**
***...and a parked mine train was derailed from the tracks.***

*Line 322 - "The last earthquake in the Hainaut coal area for which it was possible to provide a macroseismic map occurred near Carnières on 14 September 1982 at 19h24 (MW =3.4; Fig. S28). Two earthquakes were also widely felt in the region of Charleroi on 4 and 9 August 1983...". It sounds like no more earthquakes occurred after 1983? Is that really so? The authors mention something about it in this very long paper, but this point should be made very clear if we are to discuss the crucial issue of the natural vs. induced/triggered origin of this seismicity.*

**After these 1983 earthquakes, the largest events that occurred in Hainaut are three $M_L$=2.5 earthquakes, which were weakly felt. Earthquakes of magnitude below 2.0 occurred from time to time (31 earthquakes during the last 20 years), meaning that very little seismic energy was released in the coal mining area after 1985. We added this sentence at the end of *Section 2 - Earthquake Catalogue*. This seismicity is further discussed in *Section 6.4 - Focal depth determination*.**

*Line 410 - " Figure 10 reports the influence of focal depth from 1.0 to 6.0 km on the intensity attenuation curves and its stronger effect on the attenuation function than the uncertainties on the attenuation parameters. This also suggests that earthquake focal depth can be evaluated with a good accuracy using IDPs and that the differences in attenuation observed between the different earthquakes in the modelling (Fig. 9a) would likely reflect the small differences in their respective focal depths." I totally agree with this statement based on the authors' own data and based on my own experience. It is a strong conclusion that should be elucidated more extensively in the discussion.*

**Thank you for your suggestion to highlight this strong conclusion of our research even more! In the meantime, we completely rewrote the abstract to clarify all the findings and implications of our work based on the suggestions of the three reviewers. In the abstract we added the conclusion you suggested above. The current abstract reads as follows:**

*[The present study analyses the impact and damage of shallow seismic activity that occurred from the end of the 19th century until the late 20th century in the coal area of the Hainaut province in Belgium. This seismicity is the second largest source of seismic hazard in northwestern Europe, after the Lower Rhine Embayment. During this period, five earthquakes with moment magnitudes (MW) around 4.0 locally caused widespread moderate damage to buildings corresponding to maximum intensity VII in the EMS-98 scale. Reviewing intensity data from the official macroseismic surveys held by the Royal Observatory of Belgium, press reports, and contemporary scientific studies resulted in a comprehensive macroseismic intensity data set. Based on this data set, we created macroseismic maps for 28 earthquakes, and established a new Hainaut intensity attenuation model and a relationship linking magnitude, epicentral intensity and focal depth. Using these relationships, we estimated the location and magnitude of pre-1985 earthquakes that occurred prior to deployment of the modern digital Belgian Seismic network, resulting in a new updated earthquake catalogue for the Hainaut area for this period including 124 events. A comparison with other areas worldwide where currently similar shallow earthquake activity occurs, suggests that intensity attenuation is strong in Hainaut. This high attenuation and our analysis of the cumulative effect of the Hainaut seismicity indi-cate that current hazard maps overestimate ground motions in the Hainaut area. This reveals the need to use more appropriate ground motion models in hazard issues. **Another strong implication for earthquake hazard comes from the reliability of the computed focal depths that helps clarifying the hypotheses about the origin of this seismicity.** Some events were very shallow and occurred near the surface up to a depth not*

*exceeding 1 km, suggesting a close link to mining activities. Many events,including the largest shallow events in the coal area before 1970, occurred at depths greater than 2 km, which would exclude a direct relationship with mining, but still might imply a triggering causality. A similar causality can also be questioned for other events that occurred just outside of the coal area since the end of the mining works.]*

*Line 416 - I believe that Fig. 10 should be 11.*

**No, this is correct. (now Fig. 6 in the paper)**

*Line 492 - " The most destructive events occurred during or at the end of the mining exploitation." Once again, this is a crucial observation concerning the natural vs. induced nature of local seismicity. As such it should be more emphasized.*

**See our reply to line 322 and the last part of the rewritten abstract.**

*Line 518 - " All these observations suggest that the contribution of the Hainaut coal area seismicity on current seismic hazard maps in Belgium and northern France (Fig. 1) could be overestimated inside but especially outside the basin and would need to be reevaluated." This is a strong conclusion, hence I recommend the authors to clarify what exactly causes this overestimation.*

**Thank you for pointing us towards this confusion in the text. We clarified this part in this section and end the paragraph now with:**

***These observations suggest that the contribution of the Hainaut coal area seismicity to the impact of earthquake activity in southern Belgium and northern France (Fig. 1 was overestimated by comparison to consequences of other seismic sources outside the Hainaut area.***

**Furthermore, as you suggest, we repeat this strong conclusion in the *Section 7 - Conclusions* as follows:**

***[Our analysis suggests that the contribution of the Hainaut coal area seismicity on current seismic hazard maps in Belgium and northern France (Fig. 1) are overestimated and need re-evaluation, on the one hand because the magnitude of the largest events have been downsized in our new catalogue and, on the other hand, because the seismic energy is rapidly absorbed within the Hainaut coal basin due to the strong attenuation. This conclusion provides new perspectives for seismic hazard issues in Hainaut. First,....]***

*Line 614 - " Our analysis provides new perspectives for seismic hazard assessment in Hainaut by three aspects. First, it demonstrates the importance of developing a GMPE for the Hainaut area that is more in line with the observed rapid intensity decay with distance than the current existing European GMPEs". Right, but if the rapid attenuation is dominated by the shallow depth of the local seismicity, improving existing GMPEs should not be so important. The authors should clarify this point.*

**R1 asked for a comparison between the Hainaut attenuation and other regions worldwide subjected to shallow seismicity. The fact that for similar depths and magnitudes the Hainaut attenuates faster, more appropriate GMPEs that are in line with the observed rapid intensity decay with distance should be applied. Moreover, the rapid attenuation is not only a function of the shallow depth but also of the mechanical characteristics of the fractured (and slow) coal basin.**

**We clarified this in the updated conclusions.**